# Mucosal host-microbe interactions associate with clinical phenotypes in inflammatory bowel disease

Disrupted host-microbe interactions at the mucosal level are key to the pathophysiology of IBD. This study aimed to comprehensively examine crosstalk between mucosal gene expression and microbiota in patients with IBD. To study tissue-specific interactions, we perform transcriptomic (RNA-seq) and microbial (16S-rRNA-seq) profiling of 697 intestinal biopsies (645 derived from 335 patients with IBD and 52 from 16 non-IBD controls). Mucosal gene expression patterns in IBD are mainly determined by tissue location and inflammation, whereas the mucosal microbiota composition shows a high degree of individual specificity. Analysis of transcript-bacteria interactions identifies six distinct groups of inflammation-related pathways that are associated with intestinal microbiota (adjusted $P < 0.05$). An increased abundance of *Bifidobacterium* is associated with higher expression of genes involved in fatty acid metabolism, while *Bacteroides* correlates with increased metallothionein signaling. In patients with fibrostenosis, a transcriptional network dominated by immunoregulatory genes is associated with *Lachnoclostridium* bacteria in non-stenotic tissue (adjusted $P < 0.05$), while being absent in CD without fibrostenosis. In patients using TNF-α-antagonists, a transcriptional network dominated by fatty acid metabolism genes is linked to *Ruminococcaceae* (adjusted $P < 0.05$). Mucosal microbiota composition correlates with enrichment of intestinal epithelial cells, macrophages, and NK-cells. Overall, these data demonstrate the presence of context-specific mucosal host-microbe interactions in IBD, revealing significantly altered inflammation-associated gene-taxa modules, particularly in patients with fibrostenotic CD and patients using TNF-α-antagonists. This study provides compelling insights into host–microbe interactions that may guide microbiota-directed precision medicine and fuels the rationale for microbiota-targeted therapeutics as a strategy to alter disease course in IBD.

Inflammatory bowel diseases (IBD), which encompass Crohn's disease (CD) and ulcerative colitis (UC), are chronic inflammatory diseases of the gastrointestinal tract[1]. The pathogenesis of IBD is caused by a complex interplay between inherited and environmental factors, gut microbiota and the host immune system[2,3]. Alterations in gut microbiota composition and functionality are commonly observed in patients with IBD, including decreased microbial diversity, decreased abundances of butyrate-producing bacteria and increased proportions of pathobionts[3,4].

✉e-mail: r.k.weersma@umcg.nl

Disrupted host-microbe interactions are central to the pathogenesis of IBD. Relationships between host genetics and the gut microbiome have been studied in both healthy subjects and patients with IBD. For example, we previously focused on host genome–gut microbiota interactions in the context of IBD[5]. However, in order to disentangle disease mechanisms that might underlie the etiology and progression of IBD, more focus is needed on local effects, i.e. the intestinal mucosa[6]. Modulation of host mucosal gene expression by gut microbiota or effects of gene expression on microbial fitness may expose mechanisms that contribute to IBD pathogenesis, knowledge that could be utilized to explore novel therapeutic targets[7]. Most studies, however, employ fecal sampling for microbiota characterization, which precludes analysis of local interactions and their immediate impact on host intestinal expression signatures. Other studies examining mucosal gene expression–mucosal microbiome associations in the context of IBD previously identified microbial groups associated with host transcripts from immune-mediated and inflammatory pathways[7–10]. In a longitudinal study, the chemokine genes *CXCL6* and *CCL20* were negatively associated with the relative abundances of *Eubacterium rectale* and *Streptococcus*, suggesting that these bacteria are more susceptible to the actions of these chemokines[8]. Another study found an inverse association between host expression of *DUOX2*, which produces reactive oxygen species

(ROS), and the relative abundance of *Ruminococcaceae*, an association that may suggest ROS-mediated antibacterial effects[11]. However, more studies are needed to unravel IBD-associated interaction factors among mucosa-attached microbiota and host intestinal-gene expression under different conditions (e.g. inflamed vs. non-inflamed tissue) and across different patient phenotypes (e.g. disease location or medication use).

Here we analyze 697 fresh-frozen intestinal biopsies derived from 335 patients with IBD and 16 non-IBD controls, and for the same biopsies we generated both mucosal transcriptomic and microbial characterization using bulk RNA-sequencing and 16S rRNA gene sequencing, respectively. We combine both datasets and present a comprehensive investigation of mutual mucosal host-microbe interactions and by integrating these with extensive clinical characteristics available. Following this approach, we aim to not only investigate mucosal gene expressions or microbiota associations with clinical phenotypes of IBD, but also study the altered host–microbe interactions while disentangling disease-, location- and inflammation-specific associations (Fig. 1). Finally, we replicate our main results in data from an independent, publicly available cohort[8]. Such comprehensive study of mucosal host–microbe interactions could broaden our understanding of the local inflammatory responses and potentially guide microbiome-directed therapies.

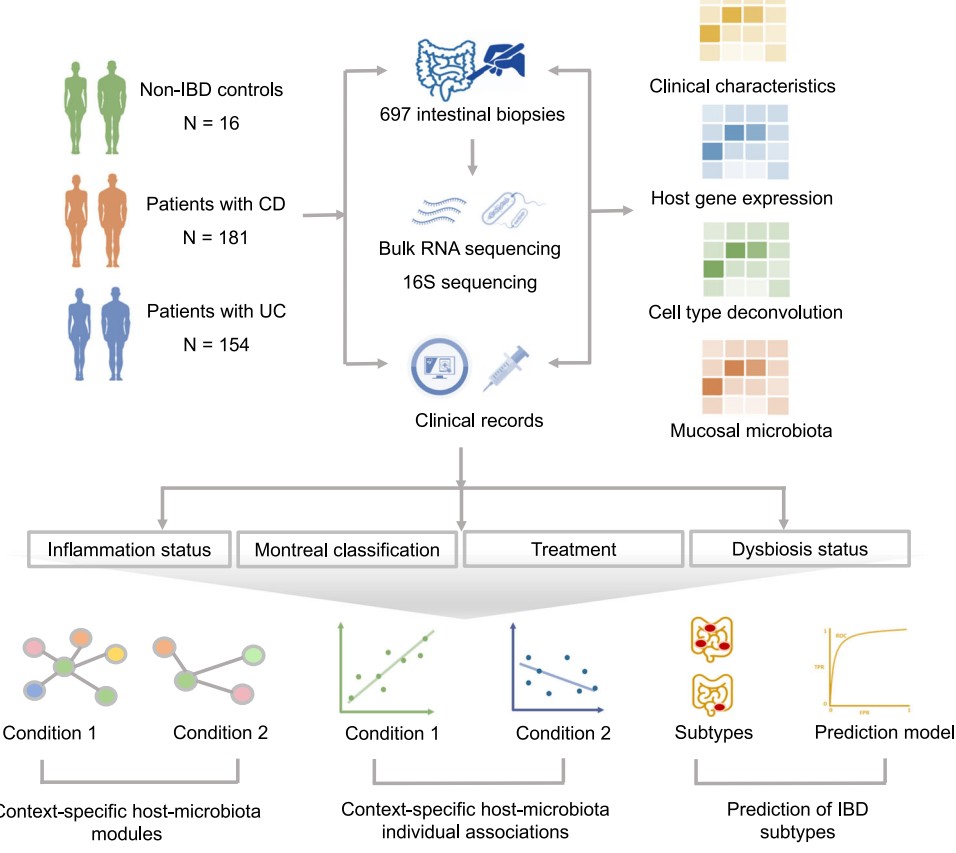

**Fig. 1 | Methodological workflow of the study.** The study cohort consisted of 335 patients with IBD (CD: $n = 181$, UC: $n = 154$) and 16 non-IBD controls, from whom 697 intestinal biopsies were collected (IBD: $n = 645$, controls: $n = 52$) and processed to perform bulk mucosal RNA-sequencing and 16S gene rRNA sequencing. Detailed phenotypic data were extracted from clinical records for all study participants. In total, 245 ileal biopsies (CD: $n = 179$, UC: $n = 57$, controls: $n = 9$) and 452 colonic biopsies (CD: $n = 177$, UC: $n = 232$, controls: $n = 43$) were included: 211 biopsies derived from inflamed regions and 434 from non-inflamed regions. Ileal biopsies from patients with UC were not included in downstream statistical analyses. Mucosal gene expression and bacterial abundances were systematically analyzed in

relation to different (clinical) phenotypes: presence of tissue inflammation, Montreal disease classification, medication use (e.g. TNF-α-antagonists) and dysbiotic status. Module-based clustering, network analysis (Sparse-CCA and centrLCC analysis) and individual pairwise gene–taxa associations were investigated to identify host–microbiota interactions in different contexts. Machine learning methods were used to predict IBD subtypes. We then analyzed the degree to which mucosal microbiota could explain the variation in intestinal cell type–enrichment (estimated by deconvolution of bulk RNA-seq data). To confirm our main findings, we used publicly available mucosal 16S and RNA-seq datasets for external validation[8].

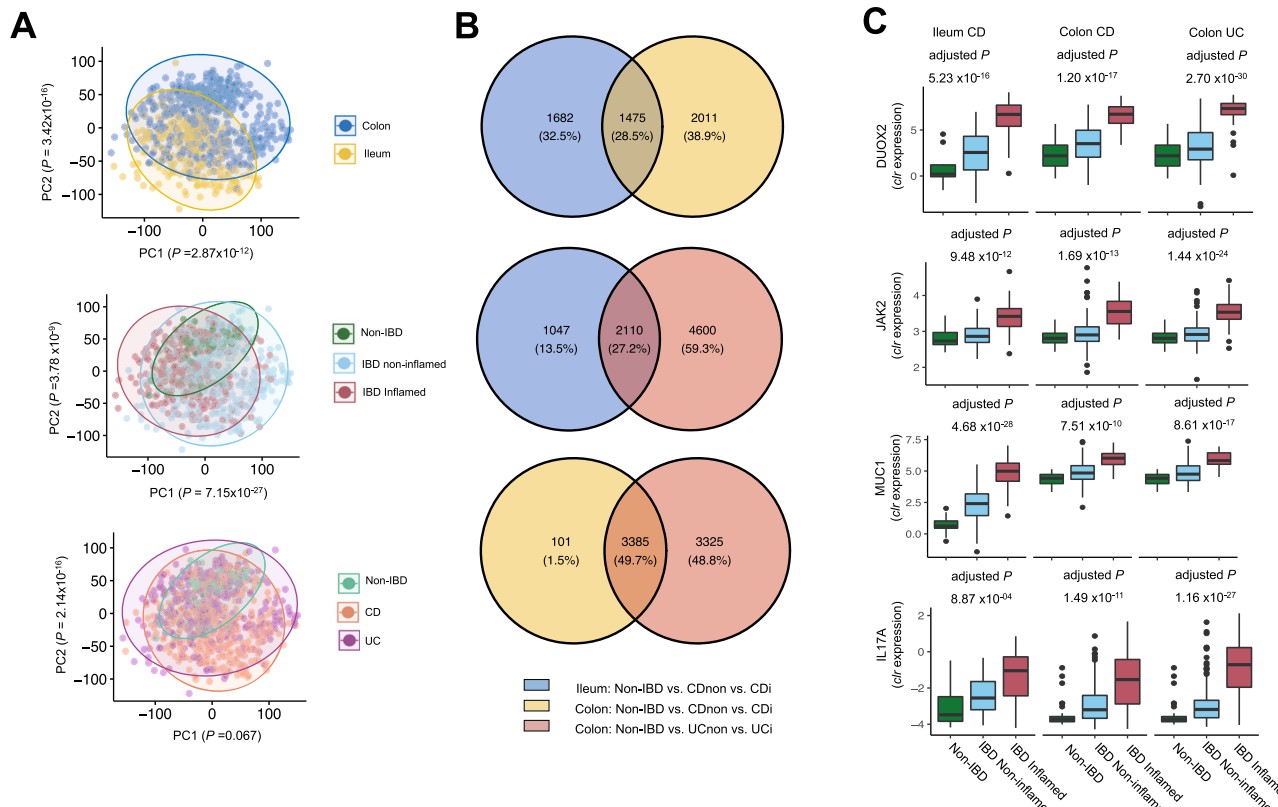

**Fig. 2 | Mucosal host gene expression patterns in intestinal tissue from patients with IBD and controls. A** Principal component analysis, labeled by tissue location (ileum/colon), inflammatory status (non-inflamed/inflamed) and disease diagnosis (control/CD/UC), shows that variation in host gene expression can be significantly explained by tissue location and inflammatory status (Wilcoxon signed-rank test). **B** Venn diagram showing the number of tissue inflammation-associated genes for all three comparisons and how many of them were shared among these comparisons: blue, ileal tissue from controls vs. non-inflamed tissue from patients with CD vs. inflamed tissue from patients with CD ($n = 3157$), yellow, colonic tissue from controls vs. non-inflamed tissue from patients with CD vs. inflamed tissue from patients with CD ($n = 3486$) and red, colonic tissue from controls vs. non-inflamed tissue from patients with UC vs. inflamed tissue from patients with UC ($n = 6710$)

(adjusted $P < 0.05$ considering multiple comparisons). **C** Relevant examples of four inflammation-associated genes, *DUOX2, JAK2, MUC1* and *IL17A*, illustrating the presence of tissue inflammation (linear regression, *t*-test, adjusted $P < 0.05$). The sample sizes from left to right are 9, 115, 65, 43, 126, 50, 43, 137 and 96. CDi, inflamed tissue from patients with Crohn's disease. CD-non, non-inflamed tissue from patients with Crohn's disease. PC, principal component. UCi, inflamed tissue from patients with ulcerative colitis. UC-non, non-inflamed tissue from patients with ulcerative colitis. Box plots show medians and the first and third quartiles (the 25th and 75th percentiles), respectively. The upper and lower whiskers extend the largest and smallest value no further than 1.5 × IQR. Source data are provided as a Source Data file.

## Results

### Cohort description

Demographic and clinical characteristics of the study population on both biopsy and patient level are provided in Supplemental Results (see Supplementary Information). In total, we included 645 intestinal biopsies from 335 patients with IBD and 52 intestinal biopsies from 16 non-IBD controls. Biopsies were derived from the colon (64.8%) and ileum (35.2%), and patients with CD and UC were similarly represented among inflamed (CD: 54.5%, UC: 45.5%) and non-inflamed (CD: 55.5%, UC: 44.5%) biopsies. The proportion of males and smokers were higher among controls (both $P < 0.01$, respectively). The remaining patient characteristics were evenly distributed among groups without significant differences, except for some types of medications which are naturally more commonly used in either patients with CD or UC.

### Mucosal gene expression reflects tissue specificity, inflammatory status and disease subtypes

First, we aimed to examine the main determinants of mucosal gene expression patterns. Principal component analysis (PCA) showed that the transcriptional patterns could be stratified by biopsy location (ileum vs. colon), inflammatory status (non-inflamed vs. inflamed) and IBD subtype (CD vs. UC) in the first two components (Fig. 2A). Tissue location and inflammatory status was significantly associated with the

first two PCs (biopsy location, ileum vs. colon: $P_{Wilcoxon} = 2.87 \times 10^{-12}$; biopsy inflammatory status, $P = 7.15 \times 10^{-27}$), whereas diagnosis (CD vs. UC vs. controls) was associated with the second PC ($P = 2.14 \times 10^{-16}$).

We then investigated dysregulated gene expressions under inflammatory status. Three differential expression comparisons between non-IBD controls, non-inflamed and inflamed biopsies stratified by diagnosis and tissue location (ileal CD, colonic CD and UC) revealed 3157, 3486, and 6710 differentially expressed genes (DEGs), respectively (adjusted $P < 0.05$) (Fig. 2B, Supplementary Data 2, Supplemental Methods). These DEGs fall mainly within interleukin signaling, neutrophil degranulation and extracellular matrix (ECM) organization pathways (adjusted $P_{Fisher} < 0.05$, Fig. S1A). In total, 1441 DEGs were identified in all three comparisons, including known upregulated genes in inflammation such as *DUOX2, MUC1, JAK2, OSM* and *IL17A* (Fig. 2C), while the down-regulated genes under inflammation were enriched in drug metabolism (Gene Set Enrichment Analysis, adjusted $P < 0.05$, Fig. S1B). We also observed an enrichment of these DEGs in IBD-associated genomic loci ($P_{Fisher} = 9.6 \times 10^{-9}$)[2].

The significant association between diagnosis and gene expression PCs suggested distinct molecular mechanisms between CD and UC (Fig. 2A). When comparing inflamed colonic tissue from patients with CD and UC, 1,466 genes were differentially abundant, of which 733 (50%) were overrepresented in CD and 733 (50%) in UC (adjusted

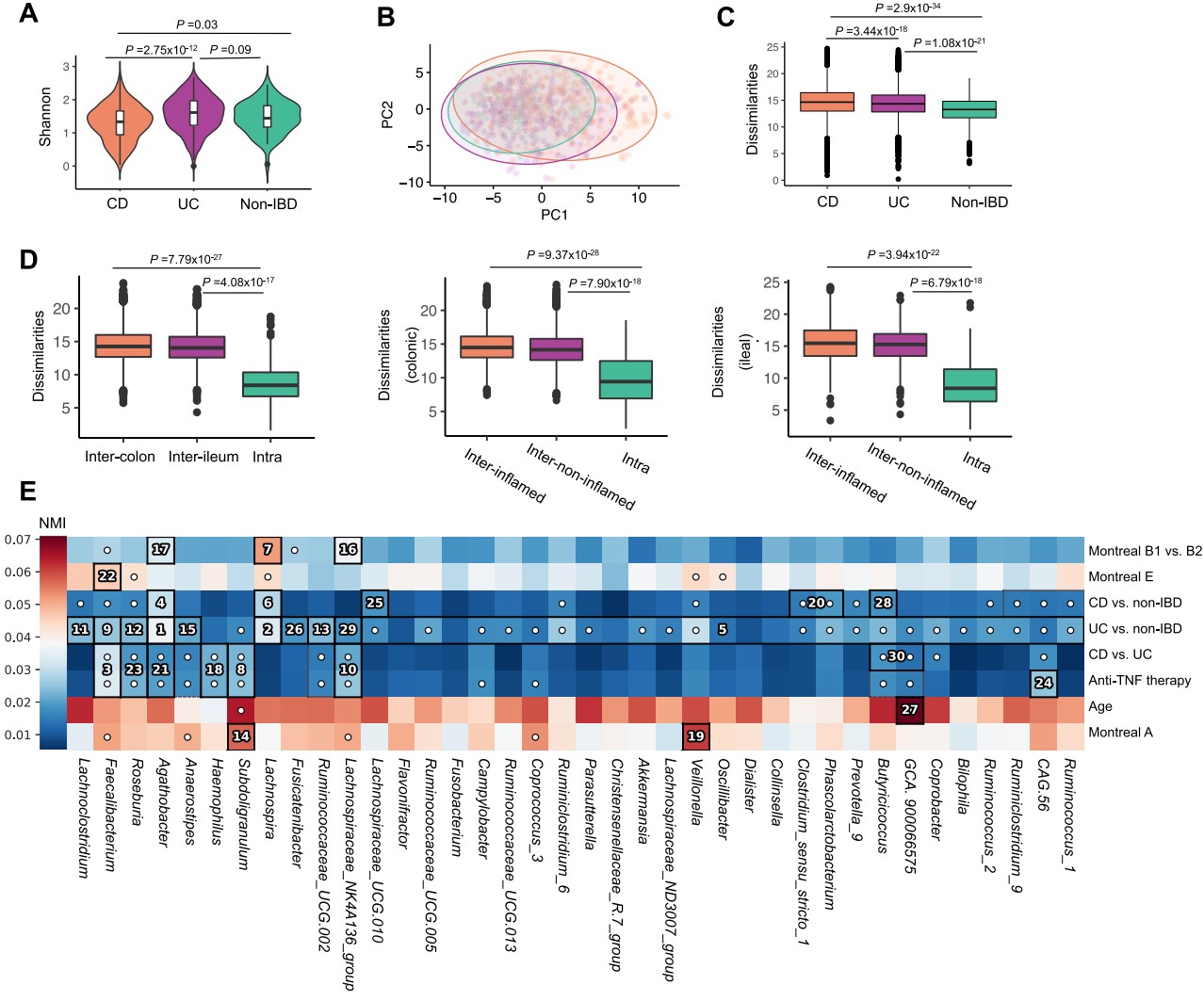

**Fig. 3 | Overall characterization of mucosa-attached microbiota in patients with IBD and controls. A** Microbial alpha-diversity (Shannon index) was lowest in samples from patients with CD (n = 356) compared to patients with UC (n = 289) and non-IBD controls (n = 52) (Wilcoxon signed-rank test). **B** PCA plot based on Aitchison's distances demonstrates the microbial dissimilarity of the mucosa-attached microbiota (colors as in **A**). **C** The degree of microbial dissimilarity (as measured by Aitchison's distances) is significantly higher in biopsies from patients with CD (n = 356), followed by patients with UC (n = 289) and non-IBD controls (n = 52) (Wilcoxon signed-rank test). **D**, Microbial dissimilarity is higher in samples from different individuals (inter-individual) when compared to paired samples from the same individual (intra-individual), which includes paired inflamed–non-inflamed tissue from ileum and colon (left panel, inter-colon: n = 11,430, inter-ileum: n = 7377, intra: n = 203), paired colonic tissue samples from inflamed and non-inflamed areas (middle panel, inter-inflamed: n = 7372, inter-non-inflamed: n = 8369, intra: n = 166) and paired ileal tissue samples from inflamed and non-

inflamed areas (right panel, inter-inflamed: n = 1590, inter-non-inflamed: n = 1592, intra: n = 73) (Wilcoxon signed-rank test). **E** Hierarchical analysis performed using an end-to-end statistical algorithm (HAllA) at genus level indicates the main phenotypic factors that correlate with intestinal mucosal microbiota composition. Significantly associated phenotypic factors were plotted after BH-approach correction. Heatmap color palette indicates the relative pairwise normalized mutual information (NMI). Numbers in cells identify significant pairs of features (phenotypic factors vs. bacterial taxa) during hierarchical analysis, where the numbers represent the descending order of statistically significant block associations based on P values in each block. White dots in cells indicate the marginal significance of a particular pair of features. CD Crohn's disease. PCA principal component analysis. UC ulcerative colitis. Box plots show medians and the first and third quartiles (the 25th and 75th percentiles), respectively. The upper and lower whiskers extend the largest and smallest value no further than 1.5 × IQR. Source data are provided as a Source Data file.

P < 0.05) (Supplementary Data 3). Pathway enrichment analysis showed the Notch-1 signaling pathway to be highly upregulated in CD compared to UC, whereas vitamin, cofactor and lipoprotein metabolism pathways were upregulated in UC (Fig. S1C)[12,13]. Thus, the underlying molecular pathways largely differed between inflamed colonic CD and UC. Subsequent cell type–deconvolution revealed that plasma cells, endothelial cells and Th2-lymphocytes were significantly increased in UC compared with CD (adjusted P < 0.05, Supplementary Data 4), further suggesting that different immunological mechanisms are involved in inflammation in CD and UC.

## Mucosal microbiota composition is highly personalized

Next, we analyzed the mucosal microbiota composition within biopsies. The most common bacterial phylum observed across all tissue samples was Bacteroidetes (CD: 58%, UC: 58%, controls: 66%), followed by Firmicutes (CD: 27%, UC: 33%, controls: 23%) and Proteobacteria (CD: 14%, UC: 8%, controls: 9%). Interestingly, across our cohort, few differentially abundant taxa were observed between colonic and ileal biopsies, and this appeared to be independent of inflammation (Fig. S2). More specifically, only seven bacterial taxa were differentially abundant between patients and controls (Supplementary Data 5, 6),

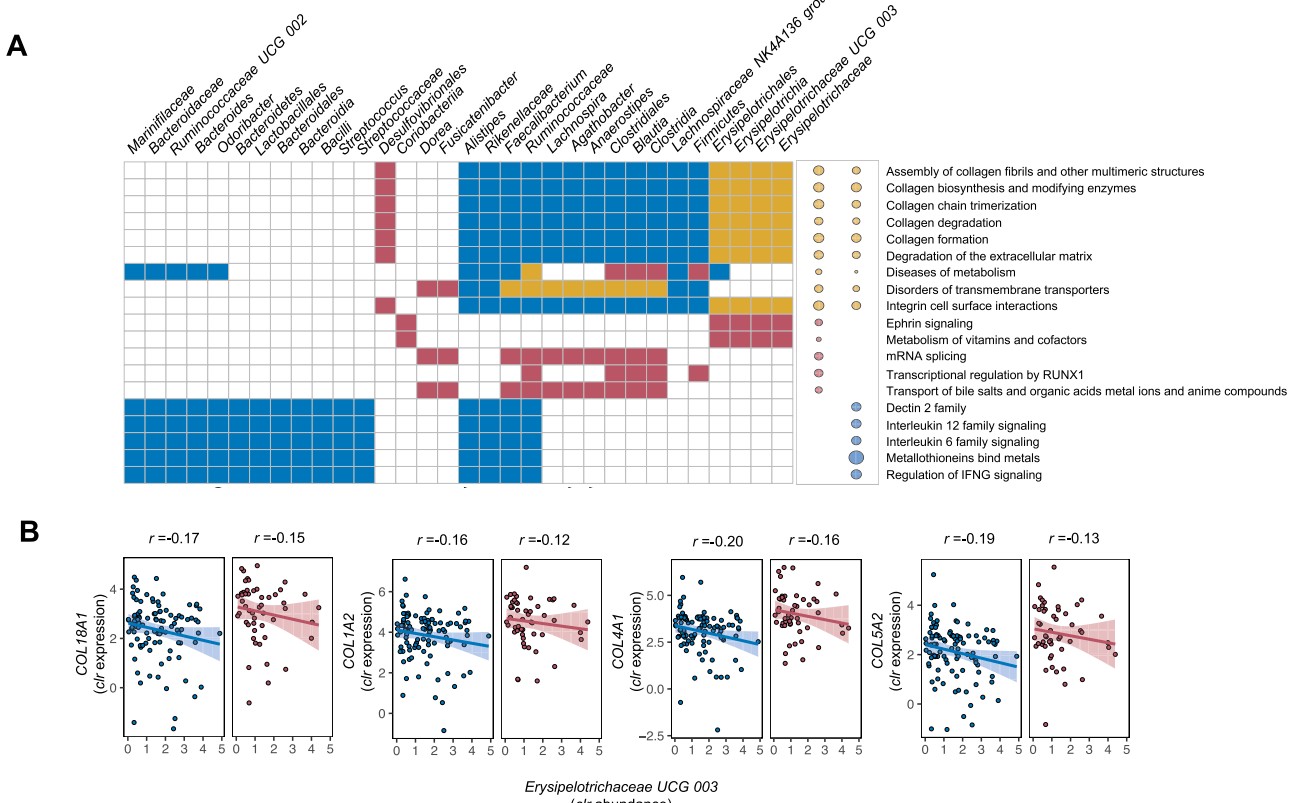

**Fig. 4 | Mucosal host–microbe interaction modules in the context of IBD.**
Sparse canonical correlation analysis (sparse-CCA) was performed across inflamed and non-inflamed biopsies to identify distinct correlation modules of mucosal gene expression vs. mucosal microbiota. Using 1441 inflammation-related genes and 131 microbial taxa as input, we identified seven distinct pairs of significantly correlated gene-microbe components in non-inflamed tissue and six distinct pairs in inflamed tissue (adjusted *P* < 0.05). **A** Heatmap showing significant component pairs from sparse-CCA analysis consisting of microbial taxa (horizontal axis) and host pathways (vertical axis) to which the involved genes were annotated (Spearman correlation, adjusted *P* < 0.05). Yellow boxes and dots indicate shared significant component pairs between inflamed and non-inflamed tissues, red colors indicate significant component pairs only in inflamed tissues, blue colors indicate significant component pairs only in non-inflamed tissues, and white colors indicate the absence of significant correlations. Dot sizes represent the degree of statistical significance of correlated component pairs. **B** Examples of inverse correlations existing between key genes involved in collagen and ECM biosynthesis (*COL18A1*, *COL1A2*, *COL4A1*, and *COL5A2*) and the mucosal abundance of *Erysipelotrichaceae UCG 003* taxon, representing the significant component pairs observed in both inflamed and non-inflamed tissues as visualized in the right upper corner of panel **A**. The shaded areas represent the 95% confidence intervals for predictions from a linear model. Source data are provided as a Source Data file.

which might however be driven by the relatively low number of non-IBD controls[8,14,15].

Shannon diversity was significantly lower in samples from patients with CD compared to UC and non-IBD controls ($P = 2.75 \times 10^{-16}$ and $P = 0.03$, respectively, Fig. 3A). This difference was still present when comparing only colonic biopsies from patients with CD to those from UC, indicating that this difference was not solely attributable to ileal CD (Fig. S3). Differences in microbial communities between tissue samples were evaluated by quantifying the Aitchison's distance (Fig. 3B, C). Similarly, in the HMP2 cohort, Shannon diversity was lowest in biopsy samples of patients with CD compared to patients with UC and non-IBD controls in data from the HMP2 cohort, alongside other comparable findings (Fig. S4)[8].

Microbial dissimilarity was lowest in paired tissue samples from the same individuals (Fig. 3D). Hierarchical clustering analysis performed on paired samples demonstrated a clear tendency of these samples to cluster together, a finding that could be well-replicated in the HMP2 data (Fig. S5A)[8]. Inter-individual variability in microbial communities was higher than that in intestinal gene expressions, confirmed by PERMANOVA analysis and merging all samples (Fig. S5B). Overall, our data demonstrate that the composition of the mucosal microbiota is highly personalized and that inter-individual variability dominates over the effects of tissue location or inflammatory status.

Similarly, inter-individual variability in mucosal microbiota composition dominated over tissue location and -inflammation effects in the HMP2 cohort (Fig. S5B)[8].

We then aimed to identify phenotypic factors associated with the individual microbiota using hierarchical association testing[16] (Fig. 3E, Supplementary Data 7). Analysis at genus level revealed that the main factors are stricturing disease in CD (fibrostenotic CD, Montreal B2), usage of TNF-α-antagonists, age at the time of sampling, age of onset and the comparisons of patients with CD vs. controls, UC vs. controls and CD vs. UC. In contrast, inflammatory status, tissue location, and disease location (according to the Montreal classification) did not show a significant effect. Similarly, age at the time of sampling, the comparisons of patients with CD vs. controls, UC vs. controls and CD vs. UC were the main factors associated with mucosal microbiota composition in the HMP2 cohort (Fig. S5C).

**Mucosal gene expression and microbiota classify IBD subtypes**
Since gene expression- and microbial differences were observed between IBD subtypes, we aimed to investigate whether combining mucosal gene expression and microbiota could predict IBD phenotypes using interpretable machine learning (XGBoost, see Methods). To avoid the effect of repeated measurements, we restricted these analyses to unique samples randomly selected from each patient. First,

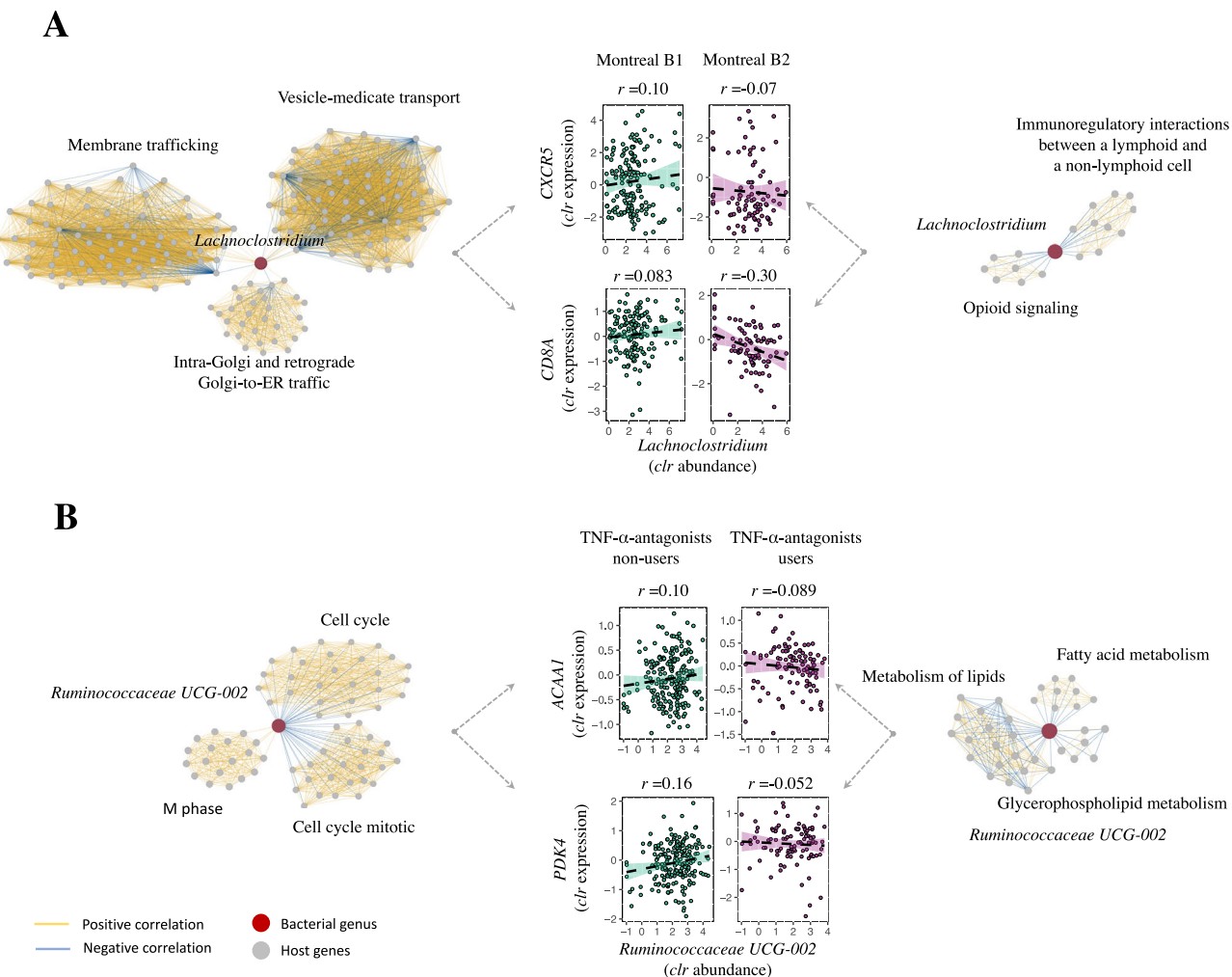

**Fig. 5 | Fibrostenotic CD and TNF-α-antagonist usage significantly alter mucosal host–microbe interactions in the context of IBD.** CentrLCC-network analyses were performed to characterize altered mucosal host–microbe interactions between different patient phenotypes. Overall, fibrostenotic CD (Montreal B2 vs. non-stricturing, non-penetrating CD, i.e. Montreal B1) and use of TNF-α-antagonists (vs. non-users) demonstrated altered interaction networks. **A** Network graphs showing an example of *Lachnoclostridium*-associated gene clusters in patients with non-stricturing, non-penetrating CD (Montreal B1) (left) and patients with fibrostenotic CD (Montreal B2) (right). *Lachnoclostridium* was the top bacteria involved (covering 65% of total associations in non-stricturing, non-penetrating CD and decreasing to 27% in fibrostenotic CD). Red dots indicate mucosal microbiota. Gray dots indicate the genes annotated by Reactome pathways. Yellow lines indicate positive associations between gene expression and bacterial abundances. Blue lines indicate negative associations. Middle panel shows key examples that significantly altered in the two patient groups, including genes involved in immunoregulatory interactions between lymphoid and non-lymphoid cells and tyrosine kinase signaling (*CD8A* and *CXCR5*). Correlations were prioritized on statistical significance. **B** Network graphs showing the example of microbiota–gene interaction networks in patients not using TNF-α-antagonists (left) vs. patients using TNF-α-antagonists (right). *Ruminococcaceae* UCG_002 was altered in interactions with host genes in patients using TNF-α-antagonists. Middle panel shows key examples of *Ruminococcaceae* UCG_002–gene interactions. These genes were involved in general biological processes such as the cell cycle but also included genes involved in fatty acid metabolism (*PDK4* and *ACAA1*). Correlations were prioritized on statistical significance. The shaded areas represent the 95% confidence intervals for predictions from a linear model. Source data are provided as a Source Data file.

we analyzed the predictive capacity of individual datasets in classifying IBD subtypes, starting with gene expressions since these showed the most prominent differences among IBD subtypes. Indeed, gene expressions showed the best predictive performance in distinguishing CD from UC compared with basic demographic factors (age, sex, BMI) and microbiota (gene expression: $AUC_{test} = 0.74$; microbial abundance: $AUC_{test} = 0.70$; basic factors: $AUC_{test} = 0.55$) (Fig. S6A). Still, the model performed better when combining gene expression and mucosal microbiota ($AUC_{test} = 0.80$). Feature attribution was quantified by SHapley Additive exPlanations (SHAP) values. The top discriminative features included the genes *SLC5A12, NPSR1, PLXDC2* and bacteria *Bilophila* and *Lachnospira*. However, Montreal classification subtypes could not be accurately predicted, neither by single data layers nor by combinations (e.g. Montreal behavior, maximum

$AUC_{test} = 0.66$; Montreal extension, maximum $AUC_{test} = 0.66$) (Fig. S6B, C, Supplementary Data 8).

## Distinct host-microbe interaction modules are identified in relation to inflammation

Tissue inflammation is one of the main contributors to intestinal DEGs (Fig. 2A, Fig. S5A). To detect how these DEGs interact with microbiota, we focused on the 1,441 DEGs associated with tissue inflammation and all the 131 microbial taxa. Sparse canonical correlation analysis (sparse-CCA) was performed using all biopsies stratified by their inflammatory status and after regressing out potential confounders, including age, sex, BMI, batch, location, and medication. In total, we found six distinct modules of genes in inflamed tissue and seven distinct modules of genes in non-inflamed tissue that were significantly correlated with

specific modules of bacterial taxa (adjusted $P < 0.05$, Supplementary Data 9-S12. To prioritize the key genes and bacteria, we performed individual pairwise gene–bacteria associations, which revealed 15 and 59 significant gene-bacteria pairs in inflamed and non-inflamed tissues, respectively (adjusted $P < 0.05$, Supplementary Data 13). Details on the most intriguing individual pairwise gene–bacteria associations are discussed in Box 1. Furthermore, we confirmed the robustness of the results was not affected by the imbalance between the sample sizes of inflamed and non-inflamed tissues through a downsampling approach (Supplemental Results).

**Mucosal *Erysipelotrichaceae* bacteria interact with collagen biosynthesis pathways.** In the significant component pair one in inflamed tissue and pair nine in non-inflamed tissue, a higher weighting of the microbial component, represented by the family *Erysipelotrichaceae*, is associated with lower expression of a wide range of ECM genes that are involved in collagen biosynthesis, integrin cell surface interactions, collagen chain trimerization, collagen fibril cross-linking, collagen fibril assembly, ECM proteoglycans and ECM/collagen degradation (Fig. 4).

**Mucosal anaerobic butyrate-producing bacteria positively correlate with transmembrane transport and inversely correlate with collagen biosynthesis.** In the significant component pair five in inflamed tissue and component pair one in non-inflamed tissue, a bacterial module mainly represented by commensal anaerobic butyrate-producing bacterial taxa including *Faecalibacterium*, *Ruminococcaceae*, *Lachnospira*, *Agathobacter*, *Blautia*, and more, is associated with genes involved in transmembrane transport. The same cluster of bacterial taxa was also associated with pathways related to mRNA splicing and transport of bile salts and organic acids in inflamed tissue, and inversely associated with collagen and ECM biosynthesis pathways in non-inflamed tissues. In addition, among inflamed tissue, the component pair ten is represented by mucosal lactic-acid-producing bacteria including *Streptococcaceae*, *Streptococcus*, *Veillonellaceae*, and Lactobacillales, which are bacteria that actively participate in physiological food digestion, particularly carbohydrate fermentation. The gene module of this component pertains to pathways mainly related to the metabolism of water-soluble vitamins and cofactors.

**Mucosal *Bacteroides* and butyrate-producing bacterial taxa associate with host interleukin signalling and metal ion response pathways.** Among non-inflamed tissue samples, the second pair of components is mainly represented by Bacteroidetes and selected taxa representing anaerobic butyrate-producing bacteria (e.g. *Alistipes*, *Faecalibacterium*, *Ruminococcaceae*). Reduced expressions of a number of interferon signaling pathways (e.g. IFN-α, IFN-β and IFN-γ as well as the IL-2, IL-4, IL-6, IL-10, and IL-12 signaling pathways) are associated with higher weighting in this microbial component. In addition, metal ion response and metallothionein (MT) pathways (e.g. metal ion transcription factors *MT1A*, *MT1E*, *MT1F*, *MT1G* and others) are positively associated with this microbial component.

**Patients with fibrostenotic CD exhibit a *Lachnoclostridium*-associated gene network involved in immune regulation**
After identifying host-microbiota modules in relation to tissue inflammation, we next wondered whether specific interaction patterns could reflect patient characteristics. Patients with fibrostenotic CD (Montreal B2, $n = 107$) and patients using TNF-α-antagonists ($n = 113$) exhibited differentially abundant microbial taxa compared to patients without fibrostenotic CD ($n = 244$) and TNF-α-antagonist non-users ($n = 583$) (Supplemental Methods), respectively. We therefore investigated intestinal gene-microbiota interactions in these phenotypes.

Pairwise comparisons between patients with non-stricturing, non-penetrating disease vs. fibrostenotic CD revealed 2639 differentially expressed genes and five differentially abundant genera (adjusted $P < 0.05$, Supplementary Data 14). When examing their associations, we observed 1405 individual gene–bacteria pairs in patients with non-stricturing, non-penetrating CD, whereas 620 individual pairs were found in patients with fibrostenotic CD (adjusted $P < 0.05$). Comparing each bacteria-associated gene cluster between patient groups (adjusted $P < 0.05$, Methods, Supplementary Data 15), we identified four distinct networks represented by mucosal *Lachnoclostridium*, *Coprococcus*, *Erysipelotrichaceae* and *Flavonifractor*. The largest network was the *Lachnoclostridium*-gene cluster, which included 907 genes in patients with non-stricturing, non-penetrating CD, and the connected genes were mainly involved in cell activation pathways such as vesicle-mediated cellular transport and membrane trafficking. In total, 166 genes were associated with *Lachnoclostridium* in patients with fibrostenotic CD (adjusted $P < 0.05$), involved in cellular immunoregulatory interactions (e.g. *CD8A*, *CLEC2B* and *CXCR5*) and opioid signaling and G alpha (s) signaling events (mediated via cAMP-dependent protein kinases, e.g. *POMC*, *GNG7* and *GNG11*) (Fig. 5A). We confirmed these analyses through downsampling the number of samples of patients with non-stricturing, non-penetrating CD to match the number of samples from patients with fibrostenosing CD (Supplemental Results). Notably, as the tissues investigated in our study were not derived from fibrotic regions, our findings show that these distinct gene-microbiota networks are already present in non-stenotic intestinal tissue.

**Use of TNF-α-antagonists is associated with *Ruminococcaceae*-associated gene interactions related to fatty acid metabolism**
Subsequently, we investigated differences in mucosal host–microbe interactions between all patients using TNF-α-antagonists and those not using TNF-α-antagonists. Pairwise comparisons revealed that TNF-α-antagonist use was significantly associated with three different bacterial taxa, *Faecalibacterium*, *Ruminococcaceae_UCG-002*, and *Ruminococcaceae_UCG-005* (all showing reduced abundances in users), and 513 different genes (adjusted $P < 0.05$, Supplementary Data 16). When examining associations, we observed 362 individual gene–bacteria pairs in patients using TNF-α-antagonists and 256 individual pairs in patients not using TNF-α-antagonists (adjusted $P < 0.05$). By comparing each taxa-associated gene cluster between patients using and not using TNF-α-antagonists, we identified a single cluster represented by mucosal *Ruminococcaceae_UCG-002* that was significantly altered in users vs. non-users (adjusted $P < 0.05$, Supplementary Data 17). *Ruminococcaceae_UCG-002* bacteria were associated with 133 genes in non-users, and these genes were mainly enriched in cell cycle–associated pathways (e.g. *PRIM1* and *PRIM2*), including mitosis-, prometaphase- and checkpoint-associated genes (Fig. 5B). However, the *Ruminococcaceae_UCG-002*-associated genes in TNF-α-antagonist users (adjusted $P < 0.05$) were predominantly involved in lipid/fatty acid metabolism (e.g. *ACAA1*, *ACSL5* and *PDK4*), glycerophospholipid biosynthesis and phospholipid metabolism. Finally, we replicated these analyses while downsampling the number of samples of patients not using TNF-α-antagonists to match the number of samples from patients using TNF-α-antagonists (Supplemental Results).

**Mucosal host-microbe interactions depend on individual dysbiotic status**
As patients with IBD suffer from microbial dysbiosis, we hypothesized that the strength and/or direction of the individual expressed gene–bacteria interactions may depend on the microbial community (eubiosis vs dysbiosis). We therefore defined dysbiosis for all intestinal biopsies based on dysbiosis scores (Fig. 6A, B, Methods). By using interaction models, 2753 individual gene–bacteria interactions showed significant alterations (interaction adjusted $P < 0.05$) (Fig. 6C,

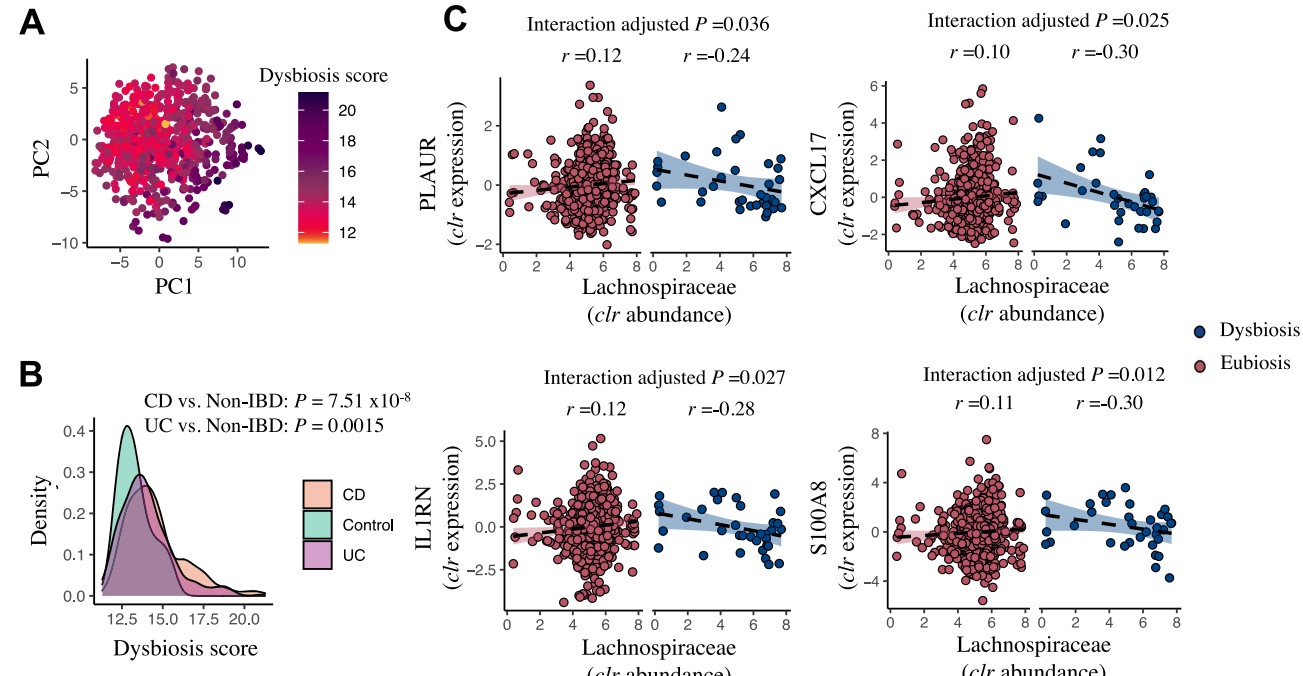

**Fig. 6 | Mucosal host−microbe interactions depend on individual dysbiotic status. A** PCA of mucosal 16 S rRNA sequencing data shows that degree of mucosal dysbiosis scores. **B** Dysbiosis scores were generally higher among patients with CD and UC compared to non-IBD controls (Spearman correlation test). **C** Key examples of individual gene−bacteria interactions that demonstrate a directional shift upon dysbiotic samples (higher dysbiosis 90−100%) as compared to patients with eubiotic samples (lower dysbiosis scores 0−90%) in IBD (linear regression model, *t* test, adjusted *P* < 0.05). Mucosal *Lachnospiraceae* bacteria positively associate with the expression of the *PLAUR, CXCL17, IL1RN* and *S100A8* genes. CD, Crohn's disease. PC, principal component. UC, ulcerative colitis. *r*, Spearman correlation coefficient. The shaded areas represent the 95% confidence intervals for predictions from a linear model. The Source data are provided as a Source Data file.

Supplementary Data 18). Permutation tests further confirmed that these interactions were not observed by chance (Methods). For example, expression of the *PLAUR* gene encoding for the urokinase plasminogen activator surface receptor was positively associated with *Lachnospiraceae* abundance, but this shifted to an inverse association when only considering dysbiotic individuals. Another example is the positive association between *S100A8*, which encodes calgranulin A, and *Lachnospiraceae*, showing a negative association in individuals with dysbiosis. Similar to the two previous examples, the observed associations between the expression of *IL1RN* (encoding for the interleukin-1 receptor antagonist protein), *CXCL17* and *Lachnospiraceae* shifted from positive to negative among patients with dysbiosis.

**Mucosal microbiota associate with variation in intestinal cell-type enrichment**

Gut barrier dysfunction and immune dysregulation are presented by altered differentiation of a wide range of intestinal cells. We, therefore, analyzed associations between mucosal microbiota and intestinal cell types (Fig. 7, Fig. S7). Deconvolution of host gene expression data revealed that mucosal microbial abundances were significantly associated with the enrichment of specific cell types, most evidently with intestinal epithelial cells, M1 macrophages, NK-cells and mucosal eosinophils. These associations appeared evident within a combination of factors potentially contributing to the explained variation in intestinal cell type enrichment, including basic factors like age, sex, and BMI, as well as medication use, inflammatory status, and tissue location (Fig. 7A). Tissue inflammatory status and location also relatively strongly contributed to the variation in most intestinal cell types. Mucosal microbiota that were significantly associated with intestinal epithelial cell enrichment typically belonged to the Firmicutes phylum, including *Agathobacter, Dialister, Lachnospira, Lachnoclostridium* and *Ruminococcaceae* (Fig. 7B, Supplementary Data 19).

## Discussion

In this study, we show distinct mucosal host−microbe interactions in intestinal tissue of patients with IBD. Mucosal gene expression patterns in IBD are mainly determined by tissue location and inflammatory status and systematically demonstrate dysregulation of distinct inflammation-associated genes, even in endoscopically non-inflamed tissue. Subsequently, we observe that the mucosal microbiota composition in patients is marked by high inter-individual variability. The main focus of our analyses, however, was to comprehensively uncover host−microbe associations specific to tissue- and patient characteristics. We identify multiple gene-taxa modules related to inflammation. Furthermore, specific interactions are significantly altered in patients with fibrostenotic CD, patients using TNF-α-antagonists and in patients with intestinal dysbiosis. Finally, we show that mucosal microbiota are significantly associated with intestinal cell type composition, in particular with epithelial cells, macrophages and NK cells.

Patients with CD and UC show differences in both intestinal gene expressions and microbial community composition. At host transcriptomic level, Notch-1 signaling pathways are upregulated in CD, while genes involved in nutrient absorption and lipid metabolism are downregulated. Activation of Notch-1 signaling has been driven by lamina propria-residing CD4+-T-lymphocytes that induce intestinal epithelial cell differentiation[12]. Notch-1 also confers protection against the development of colorectal carcinoma via p53 signaling, thereby promoting cell cycle arrests and cellular apoptosis[12,17]. Since UC patients with long-lasting colonic inflammation have a higher risk of developing IBD-associated colorectal carcinoma, we hypothesize that downregulation of Notch-1 in these patients may potentially be involved in carcinogenesis. However, this hypothesis remains speculative, since the enrichment score ratios of the genes involved in these pathways are rather small.

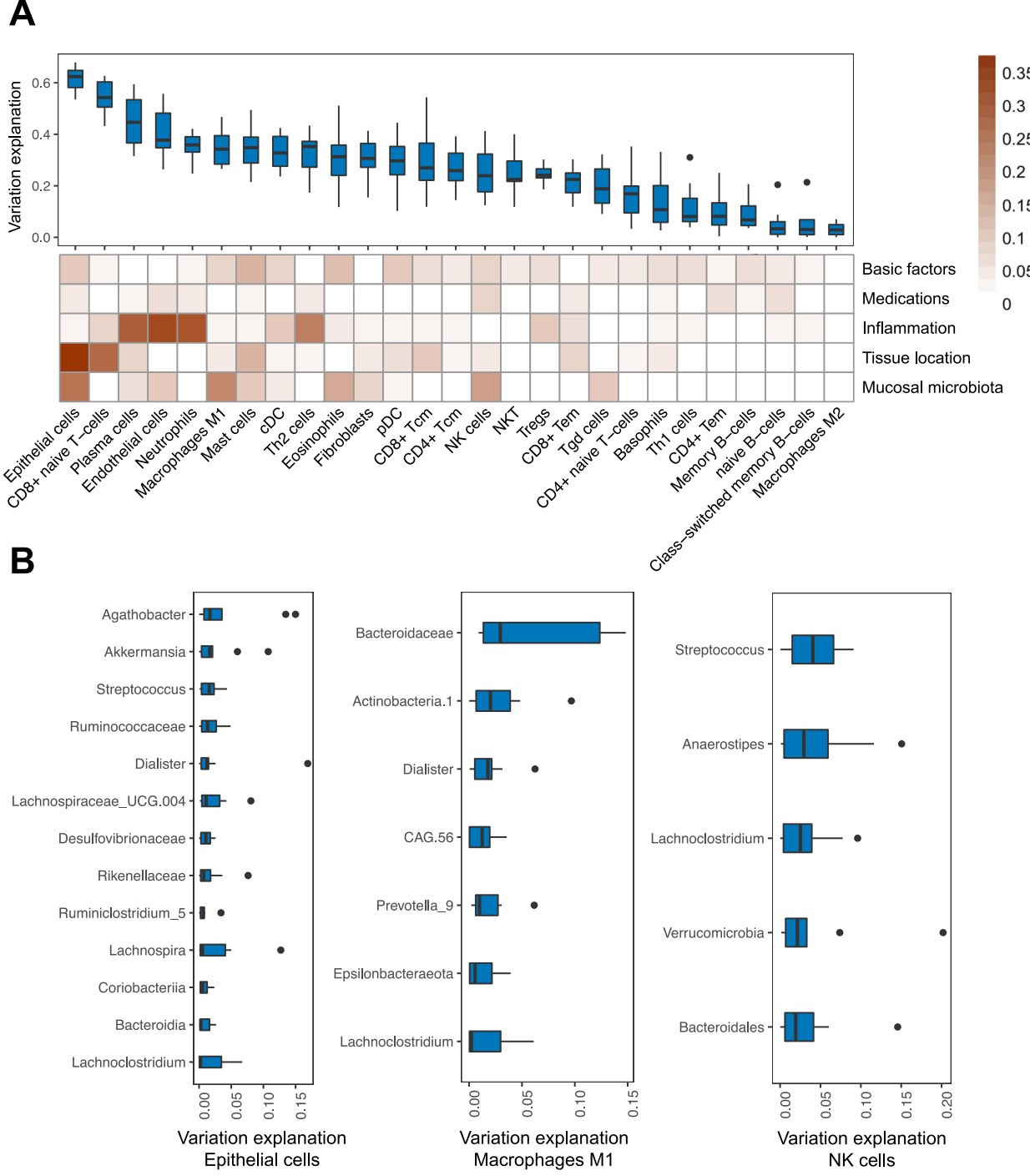

**Fig. 7 | Mucosal microbiota associate with distinct intestinal mucosal cell types. A** Boxplots show the amount of variation in intestinal cell type–enrichment that could be explained by a combination of factors. Heatmap below shows the relative contribution of different factors in explaining this intestinal cell type–enrichment, including 'basic factors' (age, sex and BMI), medication use, tissue inflammatory status, tissue location and microbiota. Mucosal microbiota contributed most to the variation in the enrichment of intestinal epithelial cells, M1-macrophages, NK cells and eosinophils. **B** Boxplots showing the contribution of the main bacterial taxa that explain the variation in mucosal enrichment of intestinal epithelial cells, M1-macrophages and NK cells—the cell types that interacted most strongly with the mucosal microbiota. Box plots show medians and the first and third quartiles (the 25th and 75th percentiles), respectively. The upper and lower whiskers extend the largest and smallest value no further than 1.5 × IQR. Source data are provided as a Source Data file.

Analysis of mucosal microbiota in patients with IBD reveals reduced alpha-diversity, microbial similarity and marked inter-individual variability that is particularly strong in CD but still present to a lesser extent in UC. These observations corroborate those of previously published mucosal 16S studies in IBD[8,14,15]. Moreover, our findings align with a recent prospective meta-analysis study that

concluded that there is a sparse evidence for microbiota-driven discrete disease subtypes within IBD[18]. Given the differences between CD and UC, integration of intestinal transcriptome and microbiota still yielded a reasonable discriminative power in this study.

IBD is characterized by heterogeneous clinical phenotypes including intestinal inflammation, disease progression (e.g.

fibrostenotic CD) and diverse treatments. We identify inflammation-associated genes in fatty acid metabolism potentially interacting with bifidobacteria, which align well with previous animal studies[19-21]. For example, treatment with *Bifidobacterium adolescentis* IM38 attenuated high fat diet–induced colitis in mice by inhibiting lipopolysaccharide production, NF-κB activation and TNF-expression in colonic epithelial cells[20]. Likewise, treatment with *Bifidobacterium infantis* ameliorated DSS-induced colitis in rats, as evidenced by decreased expression of malondialdehyde (MDA, a lipid peroxidation marker)[21]. These support the ongoing quest for efficacious probiotic (bifidobacteria-containing) supplements in patients with IBD[22,23]. We also observe that a decreased *Erysipelotrichaceae* abundance was associated with higher expressed intestinal ECM remodeling pathways. *Erysipelotrichaceae* has been associated with the expansion of mesenteric adipose tissue ("creeping fat") in CD[24]. *Erysipelotrichaceae* translocated to mesenteric fat, promoted fibrosis and stimulated tissue-remodeling, resulting in an adipose tissue barrier that may prevent systemic translocation of intestinal bacteria[24]. This phenomenon could potentially explain the negative associations between expression of ECM remodeling and *Erysipelotrichaceae*. Still, these findings should be cautiously interpreted alongside the relatively weak correlations observed between involved genes and bacterial taxa.

Another key host–microbe interaction module in relation to inflammation pertains to *Bacteroides*, which inversely (albeit weakly) correlates with interleukin signaling and positively associates with metal stress response transcription factors encoding for MTs. To maintain cellular redox balance, MTs detoxify heavy metal ions and scavenge ROS, thereby attenuating oxidative stress. Previous studies have shown that MTs may prevent experimental colitis or act as danger signals by mediating immune cell infiltration in the intestine[25,26]. Although experimental evidence seems to be inconclusive, there is ample evidence indicating a role for aberrant MT homeostasis in IBD[27]. This mechanism depends on the intracellular accumulation of zinc, which induces autophagy under chronic NOD2-stimulation. In IBD, the mucosal microbiota may contribute to the regulation of MT expression, intracellular zinc homeostasis and autophagy, thereby regulating intracellular bacterial clearance by intestinal macrophages. Findings from this study may support a putative role for *Bacteroides* in modulating MT activation[25-28]. Again, however, the findings presented surrounding this host-microbe interaction module should be cautiously interpreted given the relatively weak correlations observed between genes and bacterial taxa.

Along with the inflamed tissue effects, fibrostenotic CD and patients using TNF-α-antagonists also contribute to the change of crosstalk between host and mucosal microbiota. We observe a substantial decrease of *Lachnoclostridium*-associated genes in patients with fibrostenotic CD that mainly participate in immune system pathways. These findings suggest that *Lachnoclostridium*-associated immunoregulatory expression patterns may play a role in fibrostenotic CD. Although little is known about the exact role of *Lachnoclostridium* in IBD, these bacteria were recently strongly associated with the development of colorectal cancer and with pulmonary fibrosis[29-31]. A network of genes involved in (peroxisomal) fatty acid oxidation and lipotoxicity, shows different associations with *Ruminococcaceae* UCG-002 in patients using TNF-α-antagonists. Interestingly, multiple studies have observed that *Ruminococcaceae* increase after anti-TNF therapy in patients with IBD[32-35]. One of these studies specifically identified an association between the *Ruminococcaceae* UCG-002 group and responsiveness to TNF-α-antagonists, albeit not in relation to host gene expression patterns[33]. Strikingly, many of these genes we observe are controlled by the PPAR-γ transcription factor, a butyrate sensor that may result in reduced lipotoxicity and reduced intestinal inflammation through prevention of overgrowth of potentially pathogenic bacteria[36-38]. These findings underscore the potential relevance of PPAR-γ as a therapeutic target in IBD[38].

Disease activity in IBD could be reflected by dysbiotic status which reflects the perturbation of the intestinal microbial ecosystem[8]. By defining the patients as being eubiotic or dysbiotic, we demonstrate host–microbiota interactions that are putatively dependent on intestinal dysbiosis, including the genes involved in immunological tolerance and prevention of autoimmunity (e.g. bifidobacteria and *FOSL1/KLF2* expression), colorectal carcinogenesis (e.g. *Anaerostipes* and *SMAD4*, *Akkermansia* and *YDJC*) and inflammatory signaling (e.g. *Oscillibacter* and *OSM* expression). In addition, deconvolution of the mucosal RNA-seq data reveals cell type–specific patterns of microbial interactions that warrant further study, for example through single-cell RNA-seq studies. In this regard, our findings generated from deconvolution analysis should be carefully interpreted, since estimated cell-type fractions did not originate from non-pathological intestinal tissue and could skew the observed associations between mucosal microbiota abundances and intestinal cell-type enrichment.

Mucosal host–microbiota interactions have been investigated previously in both cohort- (e.g. the HMP2 and Irish IBD) and experimental studies[7-11]. Alongside several observations consistent with previous findings, we identify many previously unidentified host–microbe interactions. Differences in sample size, patient phenotypes and sample handling may be at least partially responsible for these observations. In our study, large groups of gene-bacteria associations are revealed that cover a wide range of molecular mechanisms potentially relevant in the context of IBD, including immune response pathways, cellular processes and a variety of metabolic pathways. Moreover, our sample size enabled us to perform an integrative analysis with respect to the large disease heterogeneity and identify previously unknown host–microbiota crosstalk related to different clinical characteristics. However, several limitations also warrant recognition. As our study is of cross-sectional nature, we cannot assess the longitudinal dynamics of host–microbe interactions to discover signatures for therapy responsiveness or disease prognosis. Consequently, our associative results cannot establish potential causality between microbial abundances and host gene expression. Additionally, there were limited numbers of non-IBD controls included in this study due to the difficulty to obtain these type of samples from healthy individuals. Therefore, our primarily focus was to unravel the heterogeneity of host-microbiota interactions among different clinical phenotypes in IBD. The 16S-sequencing for microbial characterization also has limitations when it comes to taxonomic resolution as compared to metagenomic shotgun sequencing. Another limitation pertains to the fact that we were not able to replicate all of our analyses in the validation dataset from the HMP2 cohort, since there was limited overlap in clinical phenotypes and difference in sample size (and, consequently, statistical power). Finally, bowel preparation prior to the endoscopic procedure, cross-contamination between biopsy sites during endoscopy, or differences in fecal vs. mucosal microbial profiling can affect the mucosal microbiota composition[8,14,15,39,40].

Our results demonstrate context-specific mucosal gene expressions and microbiota in IBD. Most importantly, we revealed a complex and heterogeneous interplay between host-microbiota patterns that is concomitant with the strong impact of specific IBD patient traits. Our findings may guide development of mechanistic studies (e.g. host–microbe co-culture systems) that could provide functional confirmation of relevant pathophysiological gene–bacteria interactions and serve as a resource for rational selection of therapeutic targets in IBD. This study presents a large-scale, comprehensive landscape of intestinal host–microbe interactions in IBD that could aid in guiding drug development and provide a rationale for microbiota-targeted therapy as a strategy to control disease course. For example, the gene-microbe interaction modules identified in this study may help to prioritize candidate bacterial species for pre- or probiotic modulation and could facilitate patient stratification alongside such treatment. Following this approach, therapeutics could be tailored to subgroups

of patients who would benefit most from it based on their disease characteristics. Future studies are warranted to focus on the integration of host–microbe interaction modules in prospective clinical trials investigating their utility for predicting disease course and responsiveness to treatment and for stratifying patients to facilitate therapeutic decision-making.

## Methods

### Study population

This study complies with all relevant ethical regulations and has been approved by the Institutional Review Board (IRB) of the University Medical Center Groningen (UMCG), Groningen, the Netherlands (in Dutch: 'Medisch Ethische Toetsingscommissie', METc; IRB nos. 2008/338 and 2016/424). The study was conducted according to the principles of the Declaration of Helsinki (2013). Patients with an established diagnosis of IBD were included at the outpatient clinic of the UMCG based on their participation in the 1000IBD biobank, for which detailed phenotypic information and molecular profiles have been generated[41]. Patients included in this study were at least 18 years old and were enrolled from 2003–2019. Diagnosis of IBD was based upon clinical, laboratory, endoscopic and histopathological criteria, of which the latter criteria also was used to determine the inflammatory status of collected biopsies. Detailed phenotypic data at the time of sampling were collected for all patients (Supplementary Methods). We further included 52 biopsies from 16 healthy non-IBD controls who underwent endoscopies because of clinical suspicion of intestinal disease or within the context of colon cancer screening, which were all negative. All participants provided written informed consent prior to sample collection.

### Mucosal RNA-sequencing and 16S rRNA gene sequencing

In total, 711 intestinal biopsies from 420 patients with IBD were collected. These were immediately snap-frozen in liquid nitrogen by an endoscopy nurse or research technician present during the endoscopic procedure. Biopsies from inflamed and non-inflamed areas were taken from adjacent regions, and biopsy inflammatory status was subsequently assessed histologically by certified pathologists. Biopsies were taken from ileal and colonic tissue in both patients with CD and UC, albeit inflamed ileal biopsies from patients with UC (likely due to backwash ileitis, $n = 3$) were excluded from the analyses. Biopsies were stored at −80 °C until further processing. The RNA isolation procedure and RNA-seq data processing steps are described in the Supplemental Methods. Total DNA extraction using 0.25 g of the same intestinal biopsies used for RNA sequencing was performed for 16S rRNA gene sequencing, as described previously[42–44]. Descriptions of the PCR, DNA clean-up, MiSeq library preparation, primers (Supplementary Data 1), and 16S data processing are described in the Supplemental Methods.

### Statistical analysis

**Descriptive statistics.** Descriptive data are presented as means ± standard deviation (SD), medians [interquartile range, IQR] or proportions $n$ with corresponding percentages (%). Between-group comparisons were performed using Mann-Whitney $U$-tests, Pearson's chi-squared tests or Fisher's exact tests (if $n$ observations were <10). Nominal $P$ values ≤ 0.05 were considered statistically significant.

**Mucosal gene expression analysis and microbial characterization.** Sample gene expression dissimilarity was calculated using Aitchison's distances after *clr* transformation using the R package *Compositions* (v2.02). Generalized linear mixed models were used to assess the associations between mucosal gene expression and clinical phenotypes while controlling for potential confounders, which were determined from our previous study, including age, sex, BMI, tissue location and -inflammation, medication use (aminosalicylates, thiopurines and steroids), and sample batch[45]. Moreover, the presence of multiple

biopsies per patient was accounted for by introducing a random effect in the models. Microbial richness and evenness was determined by calculating the Shannon index representing alpha-diversity of the gut microbiota. Microbial dissimilarity of samples was also determined by calculating Aitchison's distances after *clr* transformation. Analysis of paired samples from the same individuals was performed while comparing microbial features between inflammation status, disease location and disease subtype using paired Wilcoxon tests. Factors potentially influencing mucosal microbiota were determined using Hierarchical All-against-All significance testing (HAllA)[16]. A detailed description of these analyses are provided in the Supplementary Methods.

**Prediction of IBD subtypes using intestinal gene expression and microbiota.** We used eXtreme Gradient Boosting (R package *xgboost*, v1.6.0.1) method to distinguish IBD subtypes, including CD vs. UC, Montreal B1 vs B2, and Montreal E2 vs E3. The outcomes were selected based on a minimum sample size of 50. Only one unique sample was randomly selected from each individual to avoid repeated measurements effects. Each prediction model was trained in a training set (80% samples) with 5-fold cross-validation and tested in a test set (20% samples). SHapley Additive exPlanations (SHAP) values were obtained to quantify the feature contributions to the model. The following prediction models were evaluated:

 *1) IBD subtypes ~ age + gender + BMI*
 *2) IBD subtypes ~ age + gender + BMI + intestinal microbial abundance*
 *3) IBD subtypes ~ age + gender + BMI + intestinal gene expression*
 *4) IBD subtypes ~ all combined*

**Gene–microbiota interaction analysis.** We first focused on the genes that were dysregulated in inflamed (vs. non-inflamed and controls) biopsies ($n = 1441$) to investigate their potential associations with mucosal microbiota. Module-level correlations between gene expression and mucosal microbiota were performed by sparse-CCA using the residuals of genes and microbiota after correcting for age, gender, BMI, inflammation, tissue location, medication (including aminosalicylates, thiopurines and steroids), sample batch and repeated measurements separately[45]. Sparse-CCA identifies the canonical components of two paired datasets that maximizes the correlation between the relevant modules. In the sparse-CCA analyses, the lasso penalty was used to perform feature selection. The sparsity parameters (λ1, representing microbial abundance data, λ2, representing gene expression data) were tuned using a grid-search approach. More specifically, λ1 and λ2 for inflamed samples were 0.13 and 0.21, and λ1 and λ2 for non-inflamed samples were 0.10 and 0.37. The first 10 sparse-CCA components were selected and the significance was determined using the leave-one-out cross-validation approach at adjusted $P < 0.1$ (Benjamini-Hochberg, BH method). Host-enriched pathways were annotated using the Reactome database[46] for all significant components while adjusting for multiple comparisons using the BH method as implemented in the p.adjust function in R. Statistical significance was considered under an adjusted $P < 0.05$. To prioritize key gene-microbiota pairs, individual pairwise associations were assessed by fitting a generalized linear model. A gene–microbiota network analysis was visualized using the R package *ggview*.

 1) Individual gene–bacteria associations were determined using the following model:

*Gene ~ intercept + taxa + inflammation + location + age + sex + BMI + medication + batch + (1|ID)*

 Second, we focused on host–microbiota interactions associated with different clinical phenotypes (e.g., disease behavior, medication use). Genes and taxa that were differentially abundant between clinical phenotypes were selected and then served as input for CentrLCC-network analysis using the *NetCoMi* R package (v. 1.1.0). This analysis was

done in different groups separately (e.g. users and non-users of TNF-α-antagonists). To assess whether the taxa-associated gene networks were altered between groups, the associated genes for each taxa were ranked within the total geneset background based on Z-scores. The Wilcoxon test was used to compare the two gene rank lists for each taxa.

Third, we aimed to evaluate whether gene–microbiota associations would change upon the presence of intestinal dysbiosis. To do so, we modeled gene-microbiota associations using an additional interaction term in generalized linear models. IBD dysbiosis scores were presented by the median Aitchison's distances to non-IBD controls. Dysbiotic status was defined as over 90% of the scores and eubiosis was defined as being below 90% of the scores[8]. To determine whether these interactions were observed by chance, we also performed permutation tests that randomly shuffled dysbiosis 100 times across all samples, and then repeated the interaction models. On average, only three BH-adjusted significant results were obtained for each round of permutation testing, suggesting that the rate of total false positives was approximately ~ 0.014 (3/204).

2) *Gene ~ intercept + taxa + dysbiosis + taxa * dysbiosis + inflammation + location + age + sex + BMI + medication + batch + (1 | ID)*

Fourth, enrichment of specific intestinal cell types was inferred from the RNA-seq data using the *Xcell* package (v.1.1.0) in R. The effects of tissue location, inflammatory status and microbial abundances on enrichment of mucosal cell types were assessed using linear models, adjusting for age, sex, BMI, batch and medication usage. Subsequently, we used the *glmnet* R package (v.4.1.6) to investigate the variation of cell type–enrichment that could be explained by the mucosal microbiota using *lasso* regression while employing a 10-fold cross-validation using six models:

1) *Cell enrichment ~ age + gender + BMI + batch*
2) *Cell enrichment ~ medication*
3) *Cell enrichment ~ inflammation*
4) *Cell enrichment ~ tissue location*
5) *Cell enrichment ~ bacteria abundance*
6) *Cell enrichment ~ full factors mentioned above*

The percentage of explained variance ($R^2$) was calculated to estimate the variation in cell type–enrichment explained by the mucosal microbiota. All analyses were corrected for multiple comparisons using the Benjamini-Hochberg method while employing an adjusted $P$ threshold of 0.05. All gene pathway enrichment analyses were conducted using the Reactome database from MsigDB[46].

**Replication in the HMP2 dataset.** RNA-seq and 16S raw data were obtained from https://ibdmdb.org and reprocessed using the same pipeline in this study. After harmonizing with the phenotype file, we included 152 intestinal biopsies from the 85 patients with CD, 46 patients with UC and 45 non-IBD controls. Given the limited overlap in clinical phenotypes between the two cohorts, gene expression and mucosal microbiota patterns were compared separately between this study and HMP2.

**Reporting summary**
Further information on research design is available in the Nature Portfolio Reporting Summary linked to this article.

## Data availability
The datasets generated for the current study are publicly available from the European Genome-Phenome Archive (EGA) under the accession code EGAS00001002702 (mucosal RNA-seq: EGAD00001008214; 16S rRNA: EGAD00001008215). Due to patient confidentiality, the clinical data associated with the RNA-seq- and 16S rRNA-sequencing datasets are not publicly available but can be made available upon request through a minimal access procedure. This procedure consists of sending a request per email to Ms. Wieke Holwerda (w.holwerda@umcg.nl). A response will be provided within two weeks. This procedure is installed to ensure that the clinical data are being requested for scientific purposes only and thus complies with the informed consent signed by 1000IBD participants, which specifies that the collected data will not be used by commercial parties. The workflow in Fig. 1 was made in BioRender (https://www.biorender.com/). Source data are provided with this paper.

## Code availability
All analytic code used for this study can be found at the following link: https://github.com/GRONINGEN-MICROBIOME-CENTRE/Groningen-Microbiome/tree/master/Projects/IBD_biopsy_project (https://doi.org/10.5281/zenodo.10416879).

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

## Acknowledgements

All authors would like to express their gratitude towards all participants of the 1000IBD cohort. In addition, the authors would like to thank Kate McIntyre (Scientific Editor, Department of Genetics, University Medical Center Groningen) for language editing, and Ren Mao for his kind suggestions on figure styling and structural improvements (Department of Gastroenterology, The First Affiliated Hospital of Sun Yat-Sen University, Sun Yat-Sen University, Guangzhou, Guangdong, China). RKW is supported by the Seerave foundation, the Netherlands Organization for Scientific Research (NWO), and the EU Horizon Europe Program grant miGut-Health: personalized blueprint of intestinal health (101095470). SH is supported by Natural Science Foundation (NSFC) of China under grant number 82300623. ARB is supported by a Rubicon fellowship from NWO (452022317). This study was funded by Takeda Development Center Americas, Inc.

## Author contributions

Conceptualization: R.K.W. Investigation: S.H., A.R.B., R.G., B.H.J., R.M., A.B., I.J.H., J.R.B., H.J.M.H., A.V.V., L.M.S. and R.K.W. Methodology: S.H., A.R.B., R.G., B.H.J., I.J.H., J.R.B., H.J.M.H., A.V.V., L.M.S. and R.K.W. Funding acquisition: R.K.W. Supervision: E.A.M.F., A.V.V., L.M.S. and R.K.W. Writing - original manuscript: S.H., A.R.B. and L.M.S. Writing - review and editing: S.H., A.R.B., R.G., B.H.J., J.R.B., A.B., I.J.H., H.M.vD., M.C.V., K.N.F., G.D., H.J.M.H., E.A.M.F., A.V.V., L.M.S. and R.K.W.

## Competing interests

RKW acted as a consultant for Takeda, received unrestricted research grants from Takeda, Johnson & Johnson, Tramedico and Ferring and received speaker fees from MSD, Abbvie, and Janssen Pharmaceuticals. GD received an unrestricted research grant from Takeda and speaker fees from Pfizer and Janssen Pharmaceuticals. All other authors declare no competing interests.

## Additional information

Shixian Hu[1,2,3,6], Arno R. Bourgonje [1,6], Ranko Gacesa [1,2], Bernadien H. Jansen[1], Johannes R. Björk[1], Amber Bangma [1], Iwan J. Hidding[1], Hendrik M. van Dullemen[1], Marijn C. Visschedijk[1], Klaas Nico Faber[1], Gerard Dijkstra [1], Hermie J. M. Harmsen [4], Eleonora A. M. Festen[1], Arnau Vich Vila[1,2], Lieke M. Spekhorst[1,5,7] & Rinse K. Weersma [1,6] ✉

[1]Department of Gastroenterology and Hepatology, University of Groningen, University Medical Center Groningen, Groningen, the Netherlands. [2]Department of Genetics, University of Groningen, University Medical Center Groningen, Groningen, the Netherlands. [3]Institute of Precision Medicine, the First Affiliated Hospital of Sun Yat-Sen University, Sun Yat-Sen University, Guangzhou, Guangdong, China. [4]Department of Medical Microbiology, University of Groningen, University Medical Center Groningen, Groningen, the Netherlands. [5]Department of Gastroenterology and Hepatology, Medisch Spectrum Twente, Enschede, the Netherlands. [6]These authors contributed equally: Shixian Hu, Arno R. Bourgonje. [7]These authors jointly supervised this work: Lieke M. Spekhorst, Rinse K. Weersma. ✉e-mail: r.k.weersma@umcg.nl

