## [Peer Review File · Nature Communications]

Reviewers' expertise:

Reviewer #1. IBD / host gene expression / microbiome.

Reviewer #2. Microbiome / omics / bioinformatics.

Reviewer #3. IBD / microbiome / bioinformatics.

REVIEWER COMMENTS

Reviewer #1 (Remarks to the Author):

Review for nature communications - NCOMMS-23-01720

Mucosal host-microbe interactions associate with clinical phenotypes in inflammatory bowel disease

Hu et al present a study on host-microbiome interactions in IBD, which is a highly relevant topic in this research field. Most studies in the field are limited by small cohorts and thus lack statistical power. The study presented here has a powerful cohort with well selected groups, consisting of 335 IBD patients and 16 non-IBD controls. It should be appreciated that the healthy controls were not volunteers, but patients undergoing colonoscopy with no significant findings. The procedures used for sample processing document a well-established workflow, especially the immediate snap-freezing, preventing RNA degradation within the tissue. This should result in high-quality RNA-samples. The findings are based on RNA-Seq transcriptomics, 16S microbiome profiling and clinical data. Generating such a valuable resource is an achievement in itself, while the downstream analysis presented here is innovative while being focused on key questions that are currently being discussed in the research community.

While this study could contribute to the field, readers may have difficulties understanding how the authors drew their conclusions, as it is not entirely clear how the effect-sizes observed are supporting the author's key messages. This is especially relevant, as the novelty of the study is centered around the identification of specific interaction modules.

Major comments:

1. Key findings

A primary finding of the article is the interconnection between the host and the microbiome, based on a correlation analysis and interaction modules. The impact of the interaction modules is not clear. For example, IL1R2 vs. Lactobacillales: A spearman rho of 0.21 (see Fig 4A, top right panel) means that 2 out of 10 ILR2 mRNA expression values are explained by the bacterial quantities. This translates into what is called a “weak” correlation. All the modules shown have only weak correlations. How does this translate into an interpretation of the modules? Please clarify to the reader and/or describe the limitations of the modules. This is also illustrated in several figures (4, 5 and 6). In this context, the authors refer to significance of the correlation (line 235) – is this p-value corrected for multiple testing? This is very crucial, as a correlation between 131 taxa and 1441 genes results in almost 200 000 statistical tests. In case the correction was performed I suggest to i) label the p-value as “adjusted p-value” and ii) to emphasize to the reader that the interpretation is based mostly on the significance, not on the spearman rho, which corresponded to “weak” or “no correlation”. Please note that most FDR methods do not represent a correction for multiple testing. These issues have to be addressed to support the conclusions of the manuscript.

2. Deconvolution

The deconvolution employed has major limitations when interpreting patterns that do not origin from non-pathological states of tissues, thus the findings on individual cell levels have to be carefully considered -please discuss the limitation.

Minor comments:

Abstract

3. Key findings

The abstract should better cover the main message as it is presented in the discussion.

Material and methods:

4. Tissue localization:

It is unclear whether the authors did take into account the different localizations of the inflammation, as UC is mostly restricted to the colon, thus comparing the ileum of UC patients vs. CD patients may be misleading – please clarify.

5. FDR

Line 601: Which FDR Method was used? Westfall & Young? Also: Most FDR methods correct only for within-feature, not between-features, thus do not represent a multiple testing correction. Please clarify.

6. Multiple Testing

As part of this study, many different comparisons were conducted. As each additional comparison further increases the chances of finding significant differences, the results would need to be corrected for multiple comparisons - in addition to multiple testing, which needs to be performed for each comparison individually. Please add/clarify.

Results

7. Results representation in the discussion:

In general, results and figures should only be presented if they are also discussed. Please validate.

8. Figure labelling:

Many figure legends show poor labelling. Axis labelling is often unclear, units are missing, often it's unclear whether absolute or relative values are presented, normalized, scaled etc. Please update and be more precise with the labels.

9. Centroids:

In fig2 a, the centroids in the PCA plots are not very informative, in fact, for this small figure, it might be advisable to omit the centroids and show the dot clouds only.

10. Venn Diagram:

The triple three-part venn diagram is hard to read and difficult to interpret. What does it mean when a finding is significant in a 3-way comparison like non-IBD vs. CDnon-infl vs. CDinfl? Maybe split those up in multiple 2-way comparisons.

11. Ratio:

Figure 2E – it is unclear what the label “ratio” in the x-axis of the pathway plots means. One could assume it is the enrichment ratio (e.g., observed vs. expected). However, if that is the case, the ratios are extremely low: 0.01 to 0.05 – that would correspond to an enrichment of 1 to 5%. Please clarify / update the label and/or figure legend.

12. Color scale:

Figure 3e: please add a unit to the color scale legend.

13. Interaction factors:

Figure 3e: if the numbers identify significant pairs of features, the figure suggests that Agathobacter (row 1 column 4 in the heatmap) correlates to 17 phenotypic factors? Which are those? Please enable the reader to find that out by employing a supplemental table or figure.

14. Validation:

It is not clear to the reader which of the results could be validated and which couldn't be replicated due to the limitation of the validation datasets. Please emphasize/clarify this limitation in the discussion.

15. Central hub:

Figure 5a suggests that Lachnoclostridium is a central hub, connecting vesicla mediated transport, membrane trafficking and the adaptive immune system – this might be misleading (same is true for 5b). Also, there is a small typo: lachn_clostridium

16. Correlation values:

The correlations shown in figure 5 are so-called “weak” or even “no correlation” ($r=-0.06$). Moreover, some of the values origin from two distinct clouds rather than from a correlation. Please comment and address in the manuscript. Connecting two clouds oftentimes results in misleading correlations, while in fact one could be looking at two endotypes.

17. Labels and colors:

Figure 7: please add a label to the color scale in the figure. What reasoning is behind the coloring of the boxplots in panel A? Please describe in the figure legend.

18. Individual contributions vs. groups:

Figure 7: As the contribution to the variation by the individual taxa is relatively weak (or nonexistent) could it help to create functional contributor groups, illustrating a stronger effect?

Discussion

19. Individual genes:

Interpreting individual gene-findings, such as Notch-1 in the context of UC (line 420) remains speculative.

20. Effect sizes:

From line 450: if the MT findings are based on correlation coefficients as depicted in Figure 4c (around 0.05 to 0.1) then putting these weak correlations in a disease context seems quite adventurous.

21. Outlook:

From line 512: Naturally it would be desirable to employ microbiota-targeted therapies, yet the authors don't explain how the data presented can contribute to this. While this is only an outlook, the authors could at least provide a short hint how this could contribute to therapy.

Reviewer #2 (Remarks to the Author):

In this study, Su et al examine host mucosal gene expression - microbe interactions in the context of IBD. The findings are interesting and of potential significance to further mechanistic understanding of IBD. There are a few concerns that could be addressed to further improve the manuscript:

(1) It is mentioned that alpha diversity was significantly lower in CD biopsies compared to UC but there is no discussion on which microbial taxa were absent in CD.

(2.1) It is not clear to me why the overlapping set of 1441 genes for the identification of host-microbe interaction modules. The authors show that the gene expression in CD and UC is different and conclude in line 142 "molecular pathways largely differed between colonic CD and UC". I wonder if they are missing important interactions between DEGs exclusive to an IBD subtype and microbes.

(2.2) Are all of these 1441 genes truly "inflammation-linked"?

(3) The correlations between taxa and genes are poor (albeit significant) and it would be helpful to have some discussion on how biologically meaningful or reproducible these are. Perhaps a mediation analysis can be performed to confirm a few test cases?

(4) Include ref. for line 620.

(5) Indicate significance with stars in Fig. 7. Why wasn't IBD subtype included in the model?

(6) In my opinion, the manuscript is very lengthy and the main text could be shortened to only include sections most relevant to the primary question being asked.

Reviewer #3 (Remarks to the Author):

Overall comments:

In this manuscript, Hu and Bourgonje et al. performed mucosal transcriptomic and microbial profiling of 697 intestinal biopsies taken from ileum and colon of patients with IBD and non-IBD controls to characterize host-microbiota interactions in IBD. Mucosal gene expression pattern was found to be stratified by biopsy location, inflammatory status, and IBD-subtypes, whereas mucosal microbiome had high inter-individual variability. Using sparse-CCA analysis, the authors identified distinct modules of inflammation-associated genes that correlate with specific bacterial taxa. Distinct host gene-microbe association patterns were identified in patients with fibrostenotic CD, in patients using TNF- α -antagonists, and patient's dysbiosis status. Mucosal microbiota was also found to be associated with distinct intestinal cell types.

Overall, this work presents important findings about patterns of associations between host gene expression and mucosal microbiome under different conditions in patients with IBD. However, there are a few concerns regarding the analyses in the manuscript that need to be addressed:

Specific comments:

Major:

1. A sparse-CCA analysis was performed to identify host gene-microbiota associations in relation to tissue inflammation using differentially expressed genes associated with inflammation and all microbial taxa. The authors state that all biopsy samples were used in this sparse-CCA analysis. If so, how can the gene-bacteria association patterns be discerned and interpreted between the inflamed vs non-inflamed conditions, which seems like the goal of this analysis? Are the gene-bacteria associations shown in Figure 4 representative of inflamed status or non-inflamed status? Would it be better to do this analysis separately in inflamed and non-inflamed samples and then perform some comparative analysis to identify specific gene-bacteria association patterns under these two conditions? Additionally, key details on sparse-CCA analysis are lacking in the text: what penalty type was used (lasso, etc.), how were the sparsity parameters tuned for fitting the model, how was the significance of components determined, etc. Please explain these in text for clarity.

2. In the network analysis to identify gene-bacteria associations in patients with non-stricturing, non-penetrating CD vs patients with fibrostenotic CD (Fig 5A), there are twice the number of samples for non-fibrostenotic CD (n=244) compared to fibrostenotic CD (n=107). Does this difference in sample size (i.e. power) contribute towards identification of fewer gene-bacteria pairs in patients with fibrostenotic CD (541 pairs) vs those without fibrostenotic CD (1,508 pairs), and thus influence the

biological results for these conditions? Same concern applies to gene-bacteria analysis for patients using TNF- α -antagonists (n=113) vs those not using it (n=583). For a fair comparison, could you please repeat these analyses using downsampling for the larger group and assess if the gene-bacteria association patterns from the network analysis change and overall biological conclusions differ from the original analysis?

3. A predictive model combining both host gene expression and mucosal microbiome to classify IBD subtypes was shown to perform superior to models using individual data types. Given these two data types have very different distribution and sparsity characteristics, how were these two datasets combined in the model? Naively combining gene expression data and microbiome data in the model might bias the model to prioritize one feature type over another due to differences in their distributions. How was this accounted for? Also, in addition to including the AUC of models, please report other classification metrics including sensitivity, specificity, precision, and recall to provide a complete assessment of the classification performance.

Minor comments:

1. Lines 65-67: The sentence starting with "Such studies ..." when read after the previous sentence ("Most studies, however, employ fecal sampling ...") seems to imply that works cited [7-10] used fecal samples for microbiome characterization, which is incorrect. Please rephrase this sentence (lines 65-67) to avoid confusion.

2. It is stated multiple times that results from the IBD dataset used in this study were similar to those in HMP2 data (e.g. lines 192, 235, etc.). Please provide 1-2 examples of similarity for context as it can be hard to compare complex networks of gene-microbial associations by looking at figures.

3. Line 191-193: Could you please make the figures comparable between the dataset in this paper (Fig 3E) and HMP2 data (Fig S5C), e.g. use same ordering of taxa and variables, so that it is easier to compare the two results visually?

4. In Figure 3, please clarify in the legend what the numbers in the cells represent. Current explanation is unclear.

REVIEWER COMMENTS

Reviewer #1 (Remarks to the Author):

Review for nature communications - NCOMMS-23-01720

Mucosal host–microbe interactions associate with clinical phenotypes in inflammatory bowel disease

Hu et al present a study on host-microbiome interactions in IBD, which is a highly relevant topic in this research field. Most studies in the field are limited by small cohorts and thus lack statistical power. The study presented here has a powerful cohort with well selected groups, consisting of 335 IBD patients and 16 non-IBD controls. It should be appreciated that the healthy controls were no volunteers, but patients undergoing colonoscopy with no significant findings. The procedures used for sample processing document a well-established workflow, especially the immediate snap-freezing, preventing RNA degradation within the tissue. This should result in high-quality RNA-samples. The findings are based on RNA-Seq transcriptomics, 16S microbiome profiling and clinical data. Generating such a valuable resource is an achievement in itself, while the downstream analysis presented here is innovative while being focused on key questions that are currently being discussed in the research community.

While this study could contribute to the field, readers may have difficulties understanding how the authors drew their conclusions, as it is not entirely clear how the effect-sizes observed are supporting the author's key messages. This is especially relevant, as the novelty of the study is centered around the identification of specific interaction modules.

Authors' reply: First of all, we thank the reviewer for the effort put into reviewing our manuscript and the constructive comments that originated thereof.

Major comments:

1. Key findings

A primary finding of the article is the interconnection between the host and the microbiome, based on a correlation analysis and interaction modules. The impact of the interaction modules is not clear. For example, IL1R2 vs. Lactobacillales: A spearman rho of 0.21 (see Fig 4A, top right panel) means that 2 out of 10 ILR2 mRNA expression values are explained by the bacterial quantities. This translates into what is called a “weak” correlation. All the modules shown have only weak correlations. How does this translate into an interpretation of the modules? Please clarify to the reader and/or describe the limitations of the modules. This is also illustrated in several figures (4, 5 and 6). In this

context, the authors refer to significance of the correlation (line 235) – is this p-value corrected for multiple testing? This is very crucial, as a correlation between 131 taxa and 1441 genes results in almost 200 000 statistical tests. In case the correction was performed I suggest to i) label the p-value as “adjusted p-value” and ii) to emphasize to the reader that the interpretation is based mostly on the significance, not on the spearman rho, which corresponded to “weak” or “no correlation”. Please note that most FDR methods do not represent a correction for multiple testing. These issues have to be addressed to support the conclusions of the manuscript.

Authors' reply: We agree with the reviewers that these are important considerations that need to be clearly addressed in the manuscript in order to support its conclusions. In the original version, for the association analysis between 1441 genes and 131 taxa, we have indeed adopted multiple testing correction. To clarify this, in the submitted version, we emphasized the correction for multiple testing following the Benjamini-Hochberg (BH) procedure within each comparison. Now we have also taken multiple comparisons into account in each separate section of the revised manuscript. According to the reviewer's suggestions, we have labeled “p-values” as “(BH) adjusted *P* value” in these sections and changed all “FDR” values to “(BH) adjusted *P*” values. We also emphasized in the revised manuscript that interpretation was primarily prioritized on significance and not on effect sizes (lines 357-358, 363). As a complementary clarification to comment #6, we did not adopt a study-wide multiple comparison correction since each section was driven by different hypotheses (lines 663-665). In summary, we did the following to reduce the chance of false positives induced by different comparisons:

1. Section “*Distinct host-microbe interaction modules are identified in relation to inflammation*”

We performed pair-wise individual tests between 131 taxa and 1,441 genes in inflamed and non-inflamed biopsies separately, which resulted in 377,542 tests in total ($2 \times 131 \times 1,441$). We adopted the BH-adjusted *P* value for controlling for all the tests, obtaining 15 and 59 significant gene-bacteria pairs in inflamed and non-inflamed groups, respectively (adjusted p-value <0.05) (lines 243-244).

2. Section “*Patients with fibrostenotic CD exhibit a Lachnospiridium-associated gene network involved in immune regulation*”

We performed pair-wise individual tests between five taxa and 2,639 genes in non-fibrostenotic CD and fibrostenotic CD separately, which resulted in 26,390 tests in total ($2 \times 5 \times 2,639$). We adopted the BH-adjusted *P* value for controlling for all the tests, obtaining 1,405 and 620 significant gene-bacteria pairs in two groups, respectively (adjusted p-value <0.05) (lines 305-307).

3. Section “*Use of TNF- α -antagonists is associated with Ruminococcaceae-associated gene interactions related to fatty acid metabolism*”

We performed pair-wise individual tests between three taxa and 513 genes in samples from TNF- α -antagonists users and non-users separately, which resulted in 3,078 tests in total ($2*3*513$). We adopted the BH-adjusted P value for controlling for all the tests, obtaining 256 and 362 significant gene-bacteria pairs in two groups, respectively (adjusted p-value <0.05) (lines 329-332).

Following this, the reviewer is correct that our results and interpretations were prioritized on statistical significance, while not always pointing to the corresponding effect sizes of the observed associations. We have clarified this limitation in discussion (lines 454-456, 468-470)

2. Deconvolution

The deconvolution employed has major limitations when interpreting patterns that do not origin from non-pathological states of tissues, thus the findings on individual cell levels have to be carefully considered -please discuss the limitation.

Authors' reply: Indeed, this is an inherent limitation of deconvolution analysis on bulk RNA-seq data derived from diseased tissue. We have discussed this limitation in the revised manuscript.

“In this regard, our findings generated from deconvolution analysis should be carefully interpreted, since estimated cell-type fractions did not originate from non-pathological intestinal tissue and could skew the observed associations between mucosal microbiota abundances and intestinal cell-type enrichment.” (lines 496-499)

Minor comments:

Abstract

3. Key findings

The abstract should better cover the main message as it is presented in the discussion.

Authors' reply: We have attempted to better cover the main message of our study in the abstract (lines 32-34, 44-49), albeit in the end we may be constrained by the word count limits for the abstract following *Nature Communications'* author guidelines.

Material and methods:

4. Tissue localization:

It is unclear whether the authors did take into account the different localizations of the inflammation, as UC is mostly restricted to the colon, thus comparing the ileum of UC patients vs. CD patients may be misleading – please clarify.

Authors' reply: The reviewer is correct that comparing ileal biopsies from patients with CD versus those from patients with UC may provide misleading results. Although data has been generated for some ileal biopsies from patients with UC, these were not included in our analysis. We have clarified this in the Methods section and the legend of Figure 1 in the revised manuscript.

Methods section: “Biopsies were taken from ileal and colonic tissue in both patients with CD and UC, albeit inflamed ileal biopsies from patients with UC (likely due to backwash ileitis, $n=3$) were excluded from the analyses.” (lines 562-564)

Results, Figure 1: “Ileal biopsies from patients with UC were not included in downstream statistical analyses.” (lines 100)

5. FDR

Line 601: Which FDR Method was used? Westfall & Young? Also: Most FDR methods correct only for within-feature, not between-features, thus do not represent a multiple testing correction. Please clarify.

Authors' reply: The FDRs were generated following the Benjamini-Hochberg procedure, considering all features in the analyses within each hypothesis. As a complementary clarification to comment #1, we did not adopt a study-wide FDR correction (which takes all tests through the whole study into account), but instead, we have used the Benjamini-Hochberg procedure to consider all features. We have now clarified this in the Methods section of the revised manuscript:

Methods section: “Host enriched pathways were annotated using the Reactome database [46] for all significant components while adjusting for multiple comparisons using the Benjamini-Hochberg method as implemented in the `p.adjust` function in R. Statistical significance was considered under an adjusted $P < 0.05$.” (lines 620-623)

“All analyses were corrected for multiple comparisons using the Benjamini-Hochberg method while employing an adjusted P threshold of 0.05.” (lines 663-664)

Supplementary methods section “A correction for all multiple tests from three groups was applied using an BH-adjusted P threshold of 0.05.” (lines 928-929)

6. Multiple Testing

As part of this study, many different comparisons were conducted. As each additional comparison further increases the chances of finding significant differences, the results would need to be corrected for multiple comparisons - in addition to multiple testing, which needs to be performed for each comparison individually. Please add/clarify.

Authors' reply: Following the previous reply, we have clarified in the Methods section of the revised manuscript that we corrected for multiple comparisons using the Benjamini-Hochberg procedure (**lines 620-623, lines 663-664, lines 928-929**).

Results

7. Results representation in the discussion:

In general, results and figures should only be presented if they are also discussed. Please validate.

Authors' reply: We have carefully checked the results section. All figures have been addressed and discussed throughout. We further confirm that all main messages originating from each figure have also been touched upon in the Discussion section of the manuscript.

8. Figure labelling:

Many figure legends show poor labelling. Axis labelling is often unclear, units are missing, often its unclear whether absolute or relative values are presented, normalized, scaled etc. Please update and be more precise with the labels.

Authors' reply: Thank you for highlighting these issues. We carefully went over the figures and their legends to ensure all necessary details for correct interpretation including units, axes, values, and transformations are included.

9. Centroids:

In fig2 a, the centroids in the PCA plots are not very informative, in fact, for this small figure, it might be advisable to omit the centroids and show the dot clouds only.

Authors' reply: We agree with the reviewer's suggestion and the centroids have now been removed from **Figure 2A** and **Figure 3B** in the revised manuscript.

10. Venn Diagram:

The triple three-part venn diagram is hard to read and difficult to interpret. What does it mean when a finding is significant in a 3-way comparison like non-IBD vs. CDnon-infl vs. CDinfl? Maybe split those up in multiple 2-way comparisons.

Authors' reply: We thank the reviewer for raising this comment. The reason of the 3-way comparison was that we hypothesized there should be a gradient of gene expression changes ranging from non-IBD tissue, to (non-inflamed) IBD tissue, and finally to inflamed IBD tissue. Accordingly, we encoded the non-IBD samples as 0, non-inflamed IBD samples as 1 and inflamed IBD samples as 2. The gene expressions of non-inflamed IBD samples could be influenced by the sampling action since we collected these from regions adjacent to the inflamed position. By doing so, we could identify DEGs considered to be particularly inflammation-associated (adjusted P -value <0.05 , considering multiple tests and comparisons, see different trends below, x-axis indicates different comparisons and y-axis indicates the log transformed gene expressions). The three examples were selected from the total of 1,441 identified inflammation-associated genes in the main manuscript.

This analysis allowed us to further explore the potential biological roles behind DEGs between tissues. The top enriched pathways were highly consistent between comparisons from colon tissues where the DEGs fall mainly within interleukin signaling and neutrophil degranulation pathways. The top enriched pathways from ileum samples were mainly about extracellular matrix organization in addition to interleukin signaling, suggesting tissue-specific inflammation-related patterns (see figure below, also Supplementary Fig. S1).

We do agree with the reviewer, however, that the Venn diagram could be erroneously interpreted and therefore, we split the complex plot into three separate Venn diagrams. The revised **Figure 2** now looks as follows:

11. Ratio:

Figure 2E – it is unclear what the label “ratio” in the x-axis of the pathway plots means. One could assume it is the enrichment ratio (e.g., observed vs. expected). However, if that is the case, the ratios are extremely low: 0.01 to 0.05 – that would correspond to an enrichment of 1 to 5%. Please clarify / update the label and/or figure legend.

Authors' reply: Thank you for pointing out this unclarity. The enrichment significance was decided by the Fisher exact test of observed vs. expected value. The 'ratio' shown on the x-axis of these plots refers to the ratio of the number of DEGs annotated with specific pathways divided by the total number of DEGs. We have clarified its meaning in the figure legend of **Figure 2** in the revised manuscript (lines 162-163).

12. Color scale:

Figure 3e: please add a unit to the color scale legend.

Authors' reply: We apologize for the unclear color scale legend. Here we used normalized mutual information (NMI) to calculate the similarities between mixed data (e.g. categorical phenotypes, continuous bacterial data) as suggested by Ghazi *et al.*¹, (*Bioinformatics*, 2022). The color indicates the NMI value.

1. Ghazi AR, Sucipto K, Rahnavard A, Franzosa EA, McIver LJ, Lloyd-Price J, et al. High-sensitivity pattern discovery in large, paired multiomic datasets. *Bioinformatics*. 2022;38(Suppl 1):i378-i385. doi: 10.1093/bioinformatics/btac232.

13. Interaction factors:

Figure 3e: if the numbers identify significant pairs of features, the figure suggests that *Agathobacter* (row 1 column 4 in the heatmap) correlates to 17 phenotypic factors? Which are those? Please enable the reader to find that out by employing a supplemental table or figure.

Authors' reply: Indeed, the numbers identify significant pairs of features, but the numbers represent numbered block associations in descending order of statistical significance based on

P-values in each block. Each numbered block corresponds to microbial taxa co-occurring in relation to a specific phenotypic variable. A white dot indicates the marginal significance of a particular pair of features. We have clarified the meaning of the numbers and dots in the legend of **Fig. 3E** in the revised manuscript (lines 213-215). The associations between mucosal microbial taxa and phenotypic variables as visualized in **Fig. 3E** are listed in **Table S7** in the revised manuscript.

14. Validation:

It is not clear to the reader which of the results could be validated and which couldn't be replicated due to the limitation of the validation datasets. Please emphasize/clarify this limitation in the discussion.

Authors' reply: We agree this warrants clarification in the manuscript. Given the limited overlap in clinical phenotypes between our dataset and the validation dataset of the HMP2 cohort, we were forced to restrict our validation analysis to the separate validation of differential gene expression analysis and mucosal microbiota patterns. The limitations of the validation dataset, which in turn limited our replication analysis, have been emphasized in the Discussion section of the revised manuscript (lines 518-520).

15. Central hub:

Figure 5a suggests that *Lachnoclostridium* is a central hub, connecting vesicla mediated transport, membrane trafficking and the adaptive immune system – this might be misleading (same is true for 5b). Also, there is a small typo: lachn_clostridium

Authors' reply: It is indeed correct that in **Figure 5A**, the *Lachnoclostridium* taxon represented a central hub, interacting with expressed genes involved in vesicle-mediated transport, membrane trafficking and the adaptive immune system in patients with non-stricturing, non-penetrating CD. *Lachnoclostridium* was the top-associated bacterial taxon involved: it covered 65% of the total associations in patients with non-stricturing, non-penetrating CD. In **Figure 5B**, networks of *Ruminococcaceae*_UCG-002 are shown in interactions with expressed genes involved in cell cycle checkpoints and mitosis pathways, in patients not using TNF- α -antagonists. The cluster represented by *Ruminococcaceae*_UCG-002 was the only one significantly altered between users vs. non-users of TNF- α -antagonists. Albeit *Lachnoclostridium* and *Ruminococcaceae*_UCG-002 were the central bacterial taxa represented in these network analyses in **Figure 5A** and **-5B**, respectively, we do understand the reviewer's concern and modified the explanatory text in the figure legend to clarify the reason for their central positioning (lines 348-349, 358-359). In addition, we have corrected the small typos that were present in **Figure 5A**.

A**B**
16. Correlation values:

The correlations shown in figure 5 are so-called “weak” or even “no correlation” ($r=0.06$). Moreover, some of the values origin from two distinct clouds rather than from a correlation. Please comment and address in the manuscript. Connecting two clouds oftentimes results in misleading correlations, while in fact one could be looking at two endotypes.

Authors’ reply: We thank the reviewer for pointing out the “weak” or “no correlation” correlations presented in **Figure 5**. Here we performed pair-wise tests for gene-bacteria associations and compared the network difference between two conditions. For example, bacteria *Ruminococcaceae* UCG-0022 was significantly associated with *ACAA1* (adjusted $p < 0.05$, considering multiple tests and comparisons) with correlation coefficient $r=-0.13$ in TNF- α -antagonist users. However, this signal showed “weaker” or maybe even “no correlation” (adjusted $p > 0.05$) in non-users ($r=-0.06$). We admitted that $r < 0.4$ was generally considered as

“weak” correlation. We further clarified that we prioritized by statistical significance (lines 357-358, 363).

We agree with the reviewer that sometimes two clouds might lead to different correlations. In the original manuscript, we used the count zero multiplicative (CZM) method to impute zeros in the microbial data and then performed *clr* transformation. Therefore, some results might generate “two clouds” which resulted from zero-imputation values. To avoid misinterpretation in **Figure 5**, we removed those zero values and re-calculated the correlation coefficients. We found that the correlation trends were not materially changed. Only non-zero values were plotted in **Figure 5**.

17. Labels and colors:

Figure 7: please add a label to the color scale in the figure. What reasoning is behind the coloring of the boxplots in panel A? Please describe in the figure legend.

Authors' reply: There was no specific reason for the color palette applied to this figure, except for the fact that we found it pretty. Given the absence of a clear rationale for the color scale, we harmonized the color in the revised manuscript (see below for updated **Figure 7**).

A**B**
18. Individual contributions vs. groups:

Figure 7: As the contribution to the variation by the individual taxa is relatively weak (or nonexistent) could it help to create functional contributor groups, illustrating a stronger effect?

Authors' reply: We thank the reviewer for raising this interesting suggestion. We have used PICRUST2 to predict MetaCyc pathway based on 16S data¹. This resulted in 106 functional pathways (presence rate >10%). Then, we repeated the analysis to explore the bacterial functional contribution to the cell type variation. Some cell type variation indeed could be affected by microbial pathways (adjusted $P < 0.05$). For example, galacturonate pathway, which could be degraded by *E. coli* as sources of carbon for growth, explained 1% variation of epithelial cells (Figure below). However, we did not observe a stronger effect compared with individual taxa (**Figure 7, Table S19**) which is probably due to the lower resolution of amplicon-based sequencing approach^{2,3}.

Variation of cell enrichment explained by bacterial pathways predicted using PICRUST2

Cell types	Macrophages M1	NK cells	pDC	Macrophages M2	CD4+ naive T-cells	CD4+ Tcm	CD8+ naive T-cells	CD8+ Tcm	CD4+ Tem
Variation explained	0.0092	0.0290	0.0361	0.0022	0.0095	0.0077	0.0084	0.0187	0.0038

Cell types	Plasma cells	naive B-cells	Memory B-cells	Basophils	Mast cells	Endothelial cells	Epithelial cells	Fibroblasts	cDC
Variation explained	0.0144	0.0015	0.0126	0.0407	0.0054	0.0111	0.0310	0.0462	0.0581

Cell types	Tgd cells	Th2 cells	Tregs	Th1 cells	CD8+ Tem	Class-switched memory B-cells	Neutrophils	Eosinophils
Variation explained	0.0213	0.0355	0.0217	0.0063	0.0078	0.0059	0.0019	0.0027

Figure. The variation of epithelial cells, macrophages M1 and NK cells explained by Metacyc pathways.

1. Douglas, G. M., Maffei, V. J., Zaneveld, J. R., Yurgel, S. N., Brown, J. R., Taylor, C. M., ... & Langille, M. G. (2020). PICRUSt2 for prediction of metagenome functions. *Nature biotechnology*, 38(6), 685-688.
2. Lazarevic, V., Gaïa, N., Girard, M., Mauffrey, F., Ruppé, E., & Schrenzel, J. (2022). Effect of bacterial DNA enrichment on detection and quantification of bacteria in an infected tissue model by metagenomic next-generation sequencing. *ISME Communications*, 2(1), 122.
3. Cheng, W. Y., Liu, W. X., Ding, Y., Wang, G., Shi, Y., Chu, E. S., ... & Yu, J. (2022). High Sensitivity of Shotgun Metagenomic Sequencing in Colon Tissue Biopsy by Host DNA Depletion. *Genomics, Proteomics & Bioinformatics*.

Discussion

19. Individual genes:

Interpreting individual gene-findings, such as Notch-1 in the context of UC (line 420) remains speculative.

Authors' reply: We concur with the reviewer that the discussion of individual gene-taxa associations remains rather speculative and may be less biologically relevant compared to the gene-microbiota modules as identified by sparse-CCA analysis. This is also the reason why we

decided to move discussion of the top individual gene-taxa associations to supplementary boxes, for readers specifically interested in that (Box 1 and Box 2). The findings described in relation to the Notch-1 signaling pathway, however, were not based on individual genes, but on a collection of genes involved in this pathway. Still, the enrichment score ratios within these pathways are low, so there is a clear argument to downgrade the discussion of these findings. As such, we have modified the text in our discussion by stating that these findings remain speculative (lines 429-431).

20. Effect sizes:

From line 450: if the MT findings are based on correlation coefficients as depicted in Figure 4c (around 0.05 to 0.1) then putting these weak correlations in a disease context seems quite adventurous.

Authors' reply: We understand the reviewer's concern that the interpretation of these relatively weak correlations may be considered quite adventurous. Our results indeed were prioritized by statistical significance, not primarily by effect sizes. Still, the reviewer is right that our findings from the sparse-CCA analysis should be cautiously interpreted and not without considering the relatively weak correlations observed between involved genes and bacterial taxa. Therefore, we have repeatedly stated this limitation throughout the revised manuscript to remind the readers that it is important to take these effect sizes into account (lines 454-456, lines 468-470).

21. Outlook:

From line 512: Naturally it would be desirable to employ microbiota-targeted therapies, yet the authors don't explain how the data presented can contribute to this. While this is only an outlook, the authors could at least provide a short hint how this could contribute to therapy.

Authors' reply: We agree we can explain a bit more on how we think our presented data may contribute to the development of microbiota-targeted therapies. We have added a short hint to this in the final paragraph of the Discussion section in the revised manuscript (lines 532-536).

Reviewer #2 (Remarks to the Author):

In this study, Su et al examine host mucosal gene expression - microbe interactions in the context of IBD. The findings are interesting and of potential significance to further mechanistic understanding of IBD. There are a few concerns that could be addressed to further improve the manuscript:

Authors' reply: First of all, we thank the reviewer for the effort put into reviewing our manuscript and the constructive comments that originated thereof.

(1) It is mentioned that alpha diversity was significantly lower in CD biopsies compared to UC but there is no discussion on which microbial taxa were absent in CD.

Authors' reply: Indeed, we refrained from discussing differentially abundant bacterial taxa between patients and controls, or between patients with CD and UC, since in principle such differences are known from previous publications and do not substantially add to the novelty nor was the focus of our current study. For completeness, however, we have provided these data in **Tables S5-6** of the revised manuscript.

(2.1) It is not clear to me why the overlapping set of 1441 genes for the identification of host-microbe interaction modules. The authors show that the gene expression in CD and UC is different and conclude in line 142 "molecular pathways largely differed between colonic CD and UC". I wonder if they are missing important interactions between DEGs exclusive to an IBD subtype and microbes.

Authors' reply: We thank the reviewer for the constructive suggestions.

In total, 1,441 genes were selected from the overlapping results of three groups of comparisons, which should be independent of disease subtype (CD/UC) and tissue location. We defined these genes as "inflammation-associated genes". Furthermore, we focus on these genes interacting with microbiota in inflamed and non-inflamed tissues separately, and compared whether the interactions could show consistent/specific patterns. By doing so, we aimed to identify disease-relevant host-microbe interactions instead of performing a whole transcriptome-wide association analysis.

To understand the disease-specific host-microbe interactions, we have also stratified inflamed samples for colonic CD and UC and performed the sparse-CCA analysis separately in each group, the results of which are attached as supplementary file to our revised submission (**Result.sparse-CCA.CD-UC.xls**) This analysis, however, did not provide us with many IBD subtype-specific pathways nor with many shared pathways, which is likely the result of a too small sample size of these subgroups. More specifically, the only shared pathway (adjusted $P < 0.05$) found across both subgroups pertained to transmembrane transporters. In addition, one CD-specific pathway was unveiled (annotated as ion channel transport pathways) as well as one single UC-specific pathway (pertaining to glycosylation). As such, we decided to devote the main discussion of host-microbiota interaction modules to the inflammation-associated findings (see below).

(2.2) Are all of these 1441 genes truly "inflammation-linked"?

Authors' reply: Most likely, not all of these 1,441 genes are truly inflammation-linked, since other unseen factors (those we do not consider since they were not available) may impact the expression of these genes, but that is a rather generic phenomenon as observed in any comparable study. To better address this question, in the revised manuscript we present our results pertaining to the host-microbiota interaction modules as a whole as well as stratified by tissue inflammation as suggested by reviewer 3, comment 1 (see updated **Figure 4** and **lines 250-264**). In total, we identified six distinct modules of genes in non-inflamed tissue and seven modules in inflamed tissue that were significantly correlated to specific groups of bacterial taxa (adjusted $P < 0.05$). Almost all of the previously reported study population-wide modules were replicated in this stratified analysis, either in non-inflamed tissues, inflamed tissues, or both, indicating the existence of inflammation-specific host-microbiota interaction modules. For that reason, we have completely rewritten the text in the Results section to describe the main host-microbiota interaction modules detected which showed clear patterns of pathways and bacteria interacting (**lines 267-294**). The remaining modules can be found in **Tables S9-S12** alongside the revised manuscript).

(3) The correlations between taxa and genes are poor (albeit significant) and it would be helpful to have some discussion on how biologically meaningful or reproducible these are. Perhaps a mediation analysis can be performed to confirm a few test cases?

Authors' reply: We thank the reviewer for raising this suggestion. First of all, we concur with the reviewer that the correlations between genes and taxa are relatively poor in terms of effect sizes, which is actually a commonly observed phenomenon in comparable studies like ours. To confirm the associations that we have reported, we therefore performed multiple approaches, including downsampling analysis (as suggested by reviewer 3, comment 2) and permutation analyses (**lines 246-248, 316-319, 340-343, 370-371, 1150-1181**). In addition, as per suggestion of the reviewer, we performed mediation analysis to test for potential links in a few cases. We selected the top 50 differentially expressed genes associated with inflammation or fibrostenosis, to explore their mediation effects between bacteria and clinical outcomes. To avoid confounding effects, age, sex, BMI, medication, batch, tissue location and repeated measurements were regressed out for exposures and mediators separately, and the residuals were extracted for mediation analysis using the *mediation* (v.4.5) package in R (boot=T, sims=1000).

We do observe potential mediation effects of gene expressions between bacteria and certain clinical outcomes. For example, *Parasutterella* was positively associated with inflammation mediated by high expression of a group of genes including *PDK2*, *MYLIP* and *BTN3A1* ($P_{\text{mediate}} < 0.05$). Conversely, *Bifidobacterium* showed a protective effect on inflammation by suppressing genes *IL17A*, *JAK2*, *PDK2* and *DUOX2* ($P_{\text{mediate}} < 0.05$). For fibrostenosis in patients with CD, *Lachnoclostridium* might contribute to this phenotype through regulating *AK2* and *ARSA*. These two genes have been reported to be involved in liver fibrosis progression (Innes et al., *Gastroenterology* 2020). The mediation results with nominal significance at $P < 0.05$ have been attached to our revised submission (**Result.mediation.xls**). However, as our original focus was to compare host-bacteria interactions between different clinical conditions and we performed

analysis in different conditions separately, we prefer not to include the mediation results in the main manuscript at this moment.

Innes H, Buch S, Hutchinson S, et al. Genome-Wide Association Study for Alcohol-Related Cirrhosis Identifies Risk Loci in MARC1 and HNRNPUL1. *Gastroenterology*. 2020 Oct;159(4):1276-1289.e7. doi: 10.1053/j.gastro.2020.06.014.

(4) Include ref. for line 620.

Authors' reply: We have included the corresponding reference in the revised manuscript and the line number has been changed to 651:

Lloyd-Price, J. et al. Multi-omics of the gut microbial ecosystem in inflammatory bowel diseases. *Nature* **569**, 641-648 (2019). doi: 10.1038/s41586-019-1237-9.

(5) Indicate significance with stars in Fig. 7. Why wasn't IBD subtype included in the model?

Authors' reply: We apologize for the unclear description of **Figure 7**. However, there is no significance testing in this figure. To assess the different factors contributing to the cell type variations, we performed a nested 10-fold cross validation to select the best set of predictors using lasso regression based on the mean of squared errors. The white spots in **Figure 7A** indicates no variation explained by any factors. **Figure 7B** indicates the cell variation explained by bacteria as features selected by the nested cross-validation approach. Regarding IBD subtype, we did not include this in the model since we already incorporated tissue location (ileum vs. colon) and medication usage as factors. Since both variables are highly correlated with IBD subtype, we decided to refrain from adding IBD subtype as an additional variable to avoid multicollinearity issues.

(6) In my opinion, the manuscript is very lengthy and the main text could be shortened to only include sections most relevant to the primary question being asked.

Authors' reply: We fully agree with the reviewer that we produced a lengthy manuscript, which was almost inevitable given the huge amount of interesting data following the comprehensive host-microbe interaction analyses performed in this study. We did already put a lot of effort in shortening the manuscript to increase its overall readability. Many results and discussions of results have previously been moved to supplementary results and boxes. In the currently revised manuscript, we believe that the main text of the results section is succinct and factual, covering our main findings without additional unnecessary explanations. In terms of word count,

the manuscript does not exceed word limits as provided by *Nature Communications*, and, thus, is of comparable length to other published work in this field.

Reviewer #3 (Remarks to the Author):

Overall comments:

In this manuscript, Hu and Bourgonje et al. performed mucosal transcriptomic and microbial profiling of 697 intestinal biopsies taken from ileum and colon of patients with IBD and non-IBD controls to characterize host-microbiota interactions in IBD. Mucosal gene expression pattern was found to be stratified by biopsy location, inflammatory status, and IBD-subtypes, whereas mucosal microbiome had high inter-individual variability. Using sparse-CCA analysis, the authors identified distinct modules of inflammation-associated genes that correlate with specific bacterial taxa. Distinct host gene-microbe association patterns were identified in patients with fibrostenotic CD, in patients using TNF- α -antagonists, and patient's dysbiosis status. Mucosal microbiota was also found to be associated with distinct intestinal cell types.

Overall, this work presents important findings about patterns of associations between host gene expression and mucosal microbiome under different conditions in patients with IBD. However, there are a few concerns regarding the analyses in the manuscript that need to be addressed:

Authors' reply: First of all, we thank the reviewer for the positive evaluation of our manuscript, the effort put into reviewing it and the constructive comments that originated thereof.

Specific comments:

Major:

1. A sparse-CCA analysis was performed to identify host gene-microbiota associations in relation to tissue inflammation using differentially expressed genes associated with inflammation and all microbial taxa. The authors state that all biopsy samples were used in this sparse-CCA analysis. If so, how can the gene-bacteria association patterns be discerned and interpreted between the inflamed vs non-inflamed conditions, which seems like the goal of this analysis? Are the gene-bacteria associations shown in Figure 4 representative of inflamed status or non-inflamed status? Would it be better to do this analysis separately in inflamed and non-inflamed samples and then perform some comparative analysis to identify specific gene-bacteria association patterns under these two conditions? Additionally, key details on sparse-CCA analysis are lacking in the text: what penalty type was used (lasso, etc.), how were the sparsity parameters tuned for fitting the model, how was the significance of components determined, etc. Please explain these in text for clarity.

Authors' reply: We agree with the reviewer that it would make sense to stratify the sparse-CCA analysis for tissue inflammatory status. Although we lose a significant amount of power to detect the observed associations, we have performed this analysis separately and identified inflamed and non-inflamed shared or specific modules which are now presented in **Figure 4** in the revised manuscript. In total, six distinct modules of genes in non-inflamed tissue and seven in inflamed tissue were significantly correlated to specific modules of bacterial taxa (adjusted $P < 0.05$). Almost all of the previously reported study population-wide detected modules were confirmed, either in non-inflamed tissues, inflamed tissues, or both, indicating the existence of inflammation-specific host-microbiota interaction modules. Correspondingly, we have now completely rewritten the text in the Results section to describe the main host-microbiota interaction modules detected which showed a clear pattern of pathways and bacteria interacting (**lines 267-294**). The remaining modules, representing a rather 'mixed bag' of genes and taxa, can be found in **Tables S9-S12** alongside the revised manuscript).

Regarding the details surrounding the sparse-CCA analysis, we followed previously published instructions (1) on how to perform sparse-CCA in our data. The lasso penalty was used to perform feature selection. The sparsity parameters (λ_1 , microbial abundance data, λ_2 , gene expression data) were tuned using a grid-search approach. More specifically, λ_1 and λ_2 for inflamed samples were 0.13 and 0.21, and λ_1 and λ_2 for non-inflamed samples were 0.10 and 0.37. The first 10 sparse-CCA components were selected and the significance was determined using the leave-one-out cross-validation approach at adjusted $P < 0.1$. We have added the clarification in the methods section (**lines 615-620**).

(1) https://github.com/blekhmanlab/host_gene_microbiome_interactions/tree/main/Tutorial

2. In the network analysis to identify gene-bacteria associations in patients with non-stricturing, non-penetrating CD vs patients with fibrostenotic CD (Fig 5A), there are twice the number of samples for non-fibrostenotic CD (n=244) compared to fibrostenotic CD (n=107). Does this difference in sample size (i.e. power) contribute towards identification of fewer gene-bacteria pairs in patients with fibrostenotic CD (541 pairs) vs those without fibrostenotic CD (1,508 pairs), and thus influence the biological results for these conditions? Same concern applies to gene-bacteria analysis for patients using TNF- α -antagonists (n=113) vs those not using it (n=583). For a fair comparison, could you please repeat these analyses using downsampling for the larger group and assess if the gene-bacteria association patterns from the network analysis change and overall biological conclusions differ from the original analysis?

Authors' reply: We fully agree with this suggestion to avoid misleading interpretation caused by biased sample size across different groups. To assess the sample size effect on results, we have performing the following assessments:

1. *Downsampling non-inflamed tissue samples to match the number of inflamed samples*

First, we down-sampled the number of non-inflamed biopsies (n=434) 10 times to match the number inflamed biopsies (n=211), and then repeated the sparse-CCA analysis and individual

gene-bacteria associations tests with the same models and the same FDR correction method (BH). We carefully compared the results from total non-inflamed biopsies with each downsampling analysis to check the overlap in results. In general, the gene modules (adjusted $P < 0.05$) from sparse-CCA showed very good overlap rates varying from 58-71%, albeit the bacterial modules showed a bit weaker replication with rates varying from 38-69% (see **Table 1** below). Moreover, the number of significant down-sampled individual gene-bacteria results (adjusted $P < 0.05$) were lower, which were 16, 13, 9, 7, 0, 2, 15, 26, 0 and 33, but with on average a 75.86% overlap. We further checked the concordance of the 59 significant gene-bacteria associations in the total of non-inflamed biopsies in each downsampling analysis. All of them presented a good correlation (see **Figure 1** below). All extra analyses here indicate that albeit the larger sample size increases the chance to identify more significant signals, it does not substantially influence the main gene-bacteria association patterns in non-inflamed biopsies.

Table 1. Sparse-CCA results comparison of 10 times downsampling non-inflamed samples to match the inflamed samples.

Downsampling	AllSample_component	Overlap_gene_rate	Overlap_bacteria_rate
1	CP1	0.8095	0.8500
2	CP1	0.4762	0.2000
3	CP1	0.4286	0.2500
4	CP1	0.6667	0.8500
5	CP1	0.8571	0.8500
6	CP1	0.3810	0.5500
7	CP1	1.0000	0.9000
8	CP1	0.7619	0.7500
9	CP1	0.7619	0.9000
10	CP1	0.6190	0.8000
		0.6762	0.6900
1	CP2	0.8000	0.7273
2	CP2	0.8800	0.3182
3	CP2	0.1600	0.2273
4	CP2	1.0000	0.9545
5	CP2	0.9600	0.5909
6	CP2	0.9200	0.7273
7	CP2	0.4000	0.4545
8	CP2	0.8800	0.6364
9	CP2	0.6800	0.6364
10	CP2	0.6000	0.7273
		0.7009	0.6000

1	CP3	0.7917	0.7600
2	CP3	0.9583	0.2800
3	CP3	0.1667	0.2000
4	CP3	0.9167	0.8000
5	CP3	1.0000	0.9200
6	CP3	0.8750	0.7600
7	CP3	0.7500	0.2000
8	CP3	0.3750	0.5600
9	CP3	0.6250	0.8000
10	CP3	0.6667	0.8000
		0.7125	0.6080
1	CP5	0.5909	0.6333
2	CP5	0.5909	0.2333
3	CP5	0.3182	0.0667
4	CP5	0.5909	0.7333
5	CP5	0.6818	0.4000
6	CP5	0.5909	0.7333
7	CP5	0.6818	0.2000
8	CP5	0.5000	0.4333
9	CP5	0.6364	0.7000
10	CP5	0.6364	0.5000
		0.5818	0.4633
1	CP7	0.5385	0.7619
2	CP7	1.0000	0.4286
3	CP7	0.0769	0.0000
4	CP7	0.9231	0.8095
5	CP7	1.0000	0.6190
6	CP7	0.9231	0.6190
7	CP7	0.1538	0.1905
8	CP7	0.7308	0.2857
9	CP7	0.3077	0.8095
10	CP7	0.5769	0.7619
		0.6231	0.5286
1	CP8	0.6400	0.5833
2	CP8	0.8800	0.3750
3	CP8	0.0800	0.0833
4	CP8	0.9200	0.1250
5	CP8	0.9600	0.2500
6	CP8	0.9200	0.6667

7	CP8	0.6400	0.2083
8	CP8	0.0800	0.2500
9	CP8	0.6800	0.5417
10	CP8	0.4800	0.7500
		0.6280	0.3833
1	CP9	0.5217	0.8750
2	CP9	0.5652	0.2500
3	CP9	0.0870	0.2083
4	CP9	0.7391	0.5417
5	CP9	0.7826	0.6667
6	CP9	0.9565	0.5000
7	CP9	0.6957	0.2917
8	CP9	0.5652	0.5833
9	CP9	0.8261	0.7083
10	CP9	0.3913	0.9167
		0.6130	0.5542

Figure 1. Individual gene-bacteria association results with a comparison of 10 times downsampling of non-inflamed samples to match the number of inflamed samples. X-axis indicates the Z-score of gene-bacteria associations from total non-inflamed biopsies, while the Y-axis indicates the Z-score of downsampling gene-bacteria associations. *P*-values were calculated by Spearman correlation tests.

2. Downsampling samples of patients without fibrostenotic CD to match fibrostenotic CD

Subsequently, we down-sampled the amount of samples from patients without fibrostenotic CD (n=244) to match the number of samples from patients with fibrostenotic CD (n=107) for 10 times and repeated the network- and comparative analysis using the same methods. Here, we identified the same four distinct microbiota associated gene clusters between the two groups (represented by *Lachnoclostridium*, *Coprococcus*, *Erysipelotrichaceae* and *Flavonifractor*). These four clusters showed significance in 8 out of 10 times downsampling rounds (see **Table 2** below), indicating the patterns were quite stable. The *Faecalibacterium*-associated gene cluster was significant in 5 out of 10 times downsampling rounds, presumably because of a sample-size effect.

We then carefully compared the main findings of individual gene-bacteria pairs from each downsampling analysis with non-sampling results. We acknowledge that, indeed, the larger sample size increased the number of significant results, however, the main associations were very consistent across the downsampling tests. For example, the concordance of association directions was over 95% which can be observed in **Table 3** below.

Table 2. Results from downsampling analysis of network and comparative analyses regarding the presence of fibrostenotic disease.

Bacteria	Pvalue	adjusted P	Downsampling
Coprococcus_3	8.10E-28	2.45E-27	10

Coprococcus_3	2.23E-21	1.12E-20	1
Coprococcus_3	8.22E-20	4.11E-19	2
Coprococcus_3	1.32E-19	6.61E-19	8
Coprococcus_3	3.59E-19	1.80E-18	6
Coprococcus_3	6.47E-16	1.62E-15	3
Coprococcus_3	2.44E-15	1.22E-14	9
Coprococcus_3	4.90E-15	2.45E-14	7
Coprococcus_3	2.30E-12	3.84E-12	4
Coprococcus_3	6.33E-07	1.58E-06	5
Erysipelotrichaceae_UCG-003	9.79E-28	2.45E-27	10
Erysipelotrichaceae_UCG-003	2.57E-13	4.29E-13	1
Erysipelotrichaceae_UCG-003	2.04E-13	1.02E-12	5
Erysipelotrichaceae_UCG-003	5.19E-13	1.30E-12	4
Erysipelotrichaceae_UCG-003	3.76E-11	6.27E-11	6
Erysipelotrichaceae_UCG-003	4.62E-10	1.15E-09	9
Erysipelotrichaceae_UCG-003	3.39E-09	5.65E-09	2
Erysipelotrichaceae_UCG-003	3.48E-06	5.80E-06	3
Erysipelotrichaceae_UCG-003	0.356069709	0.356069709	8
Erysipelotrichaceae_UCG-003	0.56758717	0.56758717	7
Faecalibacterium	4.02E-08	4.02E-08	6
Faecalibacterium	5.65E-07	7.06E-07	4
Faecalibacterium	0.00125423	0.002090383	8
Faecalibacterium	0.022417082	0.022417082	3
Faecalibacterium	0.038724098	0.038724098	1
Faecalibacterium	0.067535932	0.067535932	2
Faecalibacterium	0.044406888	0.07401148	9
Faecalibacterium	0.065172399	0.081465498	7
Faecalibacterium	0.294672875	0.368341094	10
Faecalibacterium	0.442543424	0.442543424	5
Flavonifractor	2.77E-15	6.93E-15	1
Flavonifractor	5.72E-12	1.43E-11	7
Flavonifractor	1.59E-09	3.98E-09	8
Flavonifractor	1.61E-08	2.01E-08	6
Flavonifractor	4.70E-08	5.88E-08	2
Flavonifractor	3.47E-05	3.47E-05	4
Flavonifractor	0.002382661	0.002978327	5
Flavonifractor	0.012280824	0.01535103	3
Flavonifractor	0.329608533	0.341690345	9
Flavonifractor	0.959269783	0.959269783	10

Lachnoclostridium	2.67E-51	1.33E-50	4
Lachnoclostridium	8.48E-37	4.24E-36	3
Lachnoclostridium	6.05E-18	1.51E-17	2
Lachnoclostridium	7.12E-13	1.78E-12	6
Lachnoclostridium	3.73E-11	4.66E-11	1
Lachnoclostridium	2.19E-06	3.65E-06	10
Lachnoclostridium	1.40E-05	2.33E-05	7
Lachnoclostridium	0.000163637	0.000272728	5
Lachnoclostridium	0.040819538	0.051024423	8
Lachnoclostridium	0.341690345	0.341690345	9

Table 3. Results from downsampling analysis of individual gene-bacteria association analyses regarding the presence of fibrostenotic disease.

Bacteria	Downsampling	Individual gene-bacteria pair adjusted P replication rates	Individual gene-bacteria pair direction replication rates
Coprococcus_3	1	0.474226804	1
Coprococcus_3	2	0.298969072	1
Coprococcus_3	3	0.154639175	1
Coprococcus_3	4	0.154639175	0.989690722
Coprococcus_3	5	0.226804124	1
Coprococcus_3	6	0.164948454	1
Coprococcus_3	7	0.257731959	1
Coprococcus_3	8	0.12371134	1
Coprococcus_3	9	0.216494845	1
Coprococcus_3	10	0.453608247	1
Erysipelotrichaceae_UCG-003	1	0.046728972	0.887850467
Erysipelotrichaceae_UCG-003	2	0.247663551	1
Erysipelotrichaceae_UCG-003	3	0.397196262	1
Erysipelotrichaceae_UCG-003	4	0.121495327	0.995327103
Erysipelotrichaceae_UCG-003	5	0.098130841	0.990654206
Erysipelotrichaceae_UCG-003	6	0.08411215	0.990654206
Erysipelotrichaceae_UCG-003	7	0.275700935	1
Erysipelotrichaceae_UCG-003	8	0.53271028	0.990654206
Erysipelotrichaceae_UCG-003	9	0.514018692	1
Erysipelotrichaceae_UCG-003	10	0.397196262	1
Flavonifractor	1	0.059602649	0.986754967
Flavonifractor	2	0.337748344	0.973509934
Flavonifractor	3	0.509933775	1

Flavonifractor	4	0.324503311	1
Flavonifractor	5	0.470198675	1
Flavonifractor	6	0.284768212	1
Flavonifractor	7	0.417218543	0.993377483
Flavonifractor	8	0.238410596	1
Flavonifractor	9	0.556291391	1
Flavonifractor	10	0.377483444	1
Lachnoclostridium	1	0.292146597	1
Lachnoclostridium	2	0.52460733	1
Lachnoclostridium	3	0.69947644	1
Lachnoclostridium	4	0.695287958	1
Lachnoclostridium	5	0.413612565	0.99895288
Lachnoclostridium	6	0.2	0.996858639
Lachnoclostridium	7	0.345549738	0.993717277
Lachnoclostridium	8	0.676439791	0.997905759
Lachnoclostridium	9	0.272251309	0.989528796
Lachnoclostridium	10	0.365445026	0.99895288

3. Downsampling samples without TNF- α -antagonists usage to match samples under the usage of TNF- α -antagonists

We down-sampled the samples from patients without TNF- α -antagonists usage (n=583) to match the number of samples from patients with TNF- α -antagonists usage (n=113) for 10 times and repeated the network and comparison analysis using the same methods. In general, only the *Ruminococcaceae*-UCG_002 associated gene cluster was still present across downsampling tests, which showed significance in 7 out of 10 times downsampling rounds. The *Faecalibacterium* and *Ruminococcaceae*_UCG-005 associated gene clusters were largely influenced by sample size (see **Table 4** below).

We then carefully compared the main findings of individual gene-bacteria pairs from each downsampling analysis with non-sampling results for *Ruminococcaceae*_UCG-002 clusters. The main associations were consistent across downsampling tests. In fact, the concordance of association directions was 100% which can be observed in **Table 5**.

Table 4. Results from downsampling analysis of network and comparative analyses regarding TNF- α -antagonists usage.

Bacteria	Pvalue	adjusted P	downsample
Faecalibacterium	0.001108208	0.001662312	10
Faecalibacterium	0.003882118	0.005823176	3

Faecalibacterium	0.015090684	0.022636026	2
Faecalibacterium	0.082723638	0.082723638	5
Faecalibacterium	0.165178037	0.247767055	9
Faecalibacterium	0.265514329	0.265514329	8
Faecalibacterium	0.286810336	0.286810336	6
Faecalibacterium	0.589274419	0.589274419	1
Faecalibacterium	0.645355555	0.719987162	4
Faecalibacterium	0.852447631	0.852447631	7
Ruminococcaceae_UCG-002	1.09E-08	3.27E-08	10
Ruminococcaceae_UCG-002	2.64E-06	7.92E-06	3
Ruminococcaceae_UCG-002	5.52E-06	1.66E-05	2
Ruminococcaceae_UCG-002	4.89E-05	0.000146846	5
Ruminococcaceae_UCG-002	6.47E-05	0.000194001	9
Ruminococcaceae_UCG-002	0.001420547	0.004261642	6
Ruminococcaceae_UCG-002	0.002929913	0.008789739	1
Ruminococcaceae_UCG-002	0.023773422	0.071320265	8
Ruminococcaceae_UCG-002	0.149526134	0.224289201	7
Ruminococcaceae_UCG-002	0.09690659	0.290719771	4
Ruminococcaceae_UCG-005	0.002268538	0.003402807	5
Ruminococcaceae_UCG-005	0.013829763	0.013829763	3
Ruminococcaceae_UCG-005	0.016407254	0.016407254	10
Ruminococcaceae_UCG-005	0.02156847	0.032352706	1
Ruminococcaceae_UCG-005	0.079563424	0.119345135	6
Ruminococcaceae_UCG-005	0.190929521	0.190929521	2
Ruminococcaceae_UCG-005	0.082859946	0.224289201	7
Ruminococcaceae_UCG-005	0.213265387	0.265514329	8
Ruminococcaceae_UCG-005	0.594203446	0.594203446	9
Ruminococcaceae_UCG-005	0.719987162	0.719987162	4

Table 5. Results from downsampling analysis of individual gene-bacteria association analyses regarding TNF- α -antagonists usage.

Bacteria	Downsampling	Individual gene-bacteria pair adjusted P replication rates	Individual gene-bacteria pair direction replication rates
Ruminococcaceae_UCG-002	10	0.8	1
Ruminococcaceae_UCG-002	1	0.851851852	1
Ruminococcaceae_UCG-002	2	0.822222222	1
Ruminococcaceae_UCG-002	3	0.962962963	1
Ruminococcaceae_UCG-002	4	0.533333333	1

Ruminococcaceae_UCG-002	5	0.755555556	1
Ruminococcaceae_UCG-002	6	0.703703704	1
Ruminococcaceae_UCG-002	7	0.333333333	1
Ruminococcaceae_UCG-002	8	0.6	1
Ruminococcaceae_UCG-002	9	0.748148148	1

Overall, the larger sample size indeed elevated the chance to identify more significant signals, however, the results shown above demonstrate a substantial overlap between downsampling tests and the originally performed tests. We noticed that some of the gene-bacteria association results were influenced by the sample size, for example, the difference of *Faecalibacterium*-associated gene clusters was not significant anymore between TNF- α -antagonists users and non-users in downsampling analysis. Nevertheless, we demonstrated that most of gene-bacteria association patterns were not changed from both network and individual pair levels. Therefore, we believe that the results further examined by downsampling approach were quite robust and independent of different sample sizes in the current study. We have incorporated these findings in the main text and in the Supplementary Results of the revised manuscript (lines 246-248, 316-319, 340-342, 1150-1181).

3. A predictive model combining both host gene expression and mucosal microbiome to classify IBD subtypes was shown to perform superior to models using individual data types. Given these two data types have very different distribution and sparsity characteristics, how were these two datasets combined in the model? Naively combining gene expression data and microbiome data in the model might bias the model to prioritize one feature type over another due to differences in their distributions. How was this accounted for? Also, in addition to including the AUC of models, please report other classification metrics including sensitivity, specificity, precision, and recall to provide a complete assessment of the classification performance.

Authors' reply: We fully agree that the different distribution of gene expression and microbiota data could bias feature selection. As microbial data are generally sparser than gene expression data, we first removed bacteria with low-abundance rate at 1% and low-present rate at 10%. Then we used *clr* transformation for both gene expression and microbial data, followed by re-scaling using *scale* function in R.

We have included the updated classification metrics in **Table S8**.

Classify CD and UC

Diagnosis Model	Accuracy	Sensitivity	Specificity	Precision	AUC
Demographic factors	0.485	0.523	0.409	0.639	0.552
Demographic factors + Gut microbiota	0.652	0.676	0.621	0.694	0.697
Demographic factors + Gene expression	0.636	0.65	0.615	0.722	0.741
Demographic factors + Gut microbiota + Gene expression	0.727	0.75	0.7	0.75	0.804
Classify Montreal behavior					
Montreal B Model	Accuracy	Sensitivity	Specificity	Precision	AUC
Demographic factors	0.633	0.696	0.429	0.8	0.517
Demographic factors + Gut microbiota	0.633	0.765	0.462	0.65	0.612
Demographic factors + Gene expression	0.6	0.667	0.333	0.8	0.637
Demographic factors + Gut microbiota + Gene expression	0.633	0.737	0.455	0.7	0.662
Classify Montreal extension					
Montreal E Model	Accuracy	Sensitivity	Specificity	Precision	AUC
Demographic factors	0.64	0.5	0.652	0.111	0.535
Demographic factors + Gut microbiota	0.6	0.333	0.636	0.111	0.576

Demographic factors + Gene expression	0.64	0.5	0.706	0.444	0.625
Demographic factors + Gut microbiota + Gene expression	0.56	0.375	0.647	0.333	0.656

Minor comments:

1. Lines 65-67: The sentence starting with “Such studies ...” when read after the previous sentence (“Most studies, however, employ fecal sampling ...”) seems to imply that works cited [7-10] used fecal samples for microbiome characterization, which is incorrect. Please rephrase this sentence (lines 65-67) to avoid confusion.

Authors’ reply: Thank you for pointing this out, this is indeed an incorrect reference to the previous sentence. We have rephrased the sentence to avoid confusion. In the revised manuscript, the sentence now reads: “*Other studies examining mucosal gene expression–mucosal microbiome associations in the context of IBD previously identified microbial groups associated with host transcripts from immune-mediated and inflammatory pathways [7-10].*” (lines 69-71).

2. It is stated multiple times that results from the IBD dataset used in this study were similar to those in HMP2 data (e.g. lines 192, 235, etc.). Please provide 1-2 examples of similarity for context as it can be hard to compare complex networks of gene-microbial associations by looking at figures.

Authors’ reply: We agree that it would be good to provide a couple of examples of similarity for context, since otherwise readers have to go back to the supplementary figures each time to check for this similarity. As such, we have included a few examples of this in the Results section of the revised manuscript (lines 180-182, lines 190-191 and lines 198-200).

3. Line 191-193: Could you please make the figures comparable between the dataset in this paper (Fig 3E) and HMP2 data (Fig S5C), e.g. use same ordering of taxa and variables, so that it is easier to compare the two results visually?

Authors’ reply: Thank you for this comment. We have attempted to address this comment, but we found it a bit difficult to make a satisfactory compromise here. As the HALLA analysis (of which results are shown in **Figure 3E**) is a cluster-based method, it is impossible to break the clusters in the resulting heatmap. Moreover, the NMI values which quantify the cluster similarity

are not comparable across analyses since they do not represent absolute values. It is hard to harmonize this since the variables in our dataset versus those in the HMP2 dataset are also not the same.

4. In Figure 3, please clarify in the legend what the numbers in the cells represent. Current explanation is unclear.

Authors' reply: We have clarified the exact meaning of the numbers in this figure, which was created using the HAllA method. The numbers represent numbered block associations in descending order of statistical significance based on *P*-values in each block. Each numbered block corresponds to microbial taxa co-occurring in relation to a specific phenotypic variable. We have clarified the meaning of the numbers and dots in the legend of **Fig. 3E** in the revised manuscript (lines 213-215).

REVIEWERS' COMMENTS

Reviewer #1 (Remarks to the Author):

The manuscript presented is still highly relevant to the research field and the efforts put into the work by the authors are remarkable.

The thorough response to all comments and the clear display of how the comments were addressed is appreciated. From my point of view, all comments were appropriately addressed. I see no further needs for clarifications or revisions.

Reviewer #2 (Remarks to the Author):

Overall, I appreciate the clarifications and additional analyses provided by the authors and am of the opinion that the manuscript is significantly improved. I have a few, mostly minor, concerns that I am hoping are addressed:

(1) In the text accompanying Figure 2, it would be nice to include GSEA of the down-regulated genes. The pathway labels are cut off in Fig. 2D. I recommend moving this panel to SI since the scores are small (as pointed out by another reviewer and confirmed by the authors in Discussion) and the hits are really non-specific.

(2.1) Re Fig S2 and accompanying text: Legend title should be "phylum" and not "level". The presence of Cyanobacteria is odd - do the authors have an explanation for that? Also, were all reads assigned to only these 8 phyla?

(2.2) The authors need to use more specific language when describing changes in microbiota associated with CD vs UC biopsies in lines 168-174. I would also warn against broadly stating "overall mucosa-attached microbial composition was similar" when comparing at the genus level and ironically, in a subsection that's titled "composition is highly personalized".

(2.3) I like the comparison-with-HMP2 snippets that the authors have included in response to Reviewer 1. Can the authors please clarify if they used all IBD samples or the subset IBD samples with active disease (dysbiosis score). Since inflammatory response is so central to this study, the authors should ideally compare their observations against the HMP2 inflamed and non-inflamed biopsies separately.

(2.4) The p-value is really small for the CD-UC comparison in Fig. 2C where the distributions look pretty similar. How many comparisons in each distribution and what test was used? Can the authors confirm with Bray-Curtis distance?

(3) I am not quite sure how to interpret the HALLAgram (Fig. 3E). What do the authors mean by 'comparisons of patients with CD vs. UC' and how did the authors do the three diagnoses comparisons in the same model. I would also recommend using the most recent version of HALLA to reproduce these results.

(4) It would be helpful to the reader to include a legend in Fig. 4A in addition to the details provided in the caption.

(5) There is a lot of white space in Fig. 5.

(6) Maintain consistency in figures for indicating 'non-IBD'. Some legends say "non-IBD" and others say "Control".

(7) It is still not clear to me what the authors are trying to show in Fig. 7A. The y-axis needs to be more informative than "variation explanation" and the legend title is missing. Is the variation explained by something 60% for epithelial cells? What is that something? I urge the authors to add more details and explanation to the accompanying text.

Reviewer #3 (Remarks to the Author):

The authors have satisfactorily addressed the concerns raised in the previous review. Additional analyses have been performed to address questions and the manuscript has been appropriately updated.

REVIEWERS' COMMENTS

Reviewer #1 (Remarks to the Author):

The manuscript presented is still highly relevant to the research field and the efforts put into the work by the authors are remarkable.

The thorough response to all comments and the clear display of how the comments were addressed is appreciated. From my point of view, all comments were appropriately addressed. I see no further needs for clarifications or revisions.

Authors' reply: We would like to thank the reviewer for the positive evaluation of our manuscript and the valuable comments that were provided for the revision of our manuscript. We are pleased to read that our revision has been well received and look forward to seeing our manuscript appearing online.

Reviewer #2 (Remarks to the Author):

Overall, I appreciate the clarifications and additional analyses provided by the authors and am of the opinion that the manuscript is significantly improved. I have a few, mostly minor, concerns that I am hoping are addressed:

Authors' reply: We would like to thank the reviewer for the positive evaluation of our manuscript and the valuable comments that were provided for the revision of our manuscript. We are pleased to read that our revision has been well received and are happy to address the final, mostly minor, concerns raised during this second revision. Below, we have provided point-by-point clarifications to these comments.

(1) In the text accompanying Figure 2, it would be nice to include GSEA of the down-regulated genes. The pathway labels are cut off in Fig. 2D. I recommend moving this panel to SI since the scores are small (as pointed out by another reviewer and confirmed by the authors in Discussion) and the hits are really non-specific.

Authors' reply: Indeed, we agree that the strength of the enrichment ratios is not particularly great, although we do think that these signals confer biological relevance in the context of both CD and UC (see lines 144-147, 441-451). We followed the reviewer's recommendation and moved panels D and E from Figure 2 to the SI (Fig. S2C). Furthermore,

we have modified the main text accompanying Figure 2 as following which is also updated in Figure. S2B:

“... while the down-regulated genes under inflammation were enriched in drug metabolism (Gene Set Enrichment Analysis, adjusted $P < 0.05$)”

Regarding GSEA of downregulated genes, we decided not to pursue this, since this would yield the same view of the differentially enriched pathways between CD and UC (i.e., downregulated genes/pathways in CD are the ones being upregulated in UC and vice versa).

(2.1) Re Fig S2 and accompanying text: Legend title should be "phylum" and not "level".

The presence of Cyanobacteria is odd - do the authors have an explanation for that? Also, were all reads assigned to only these 8 phyla?

Authors' reply: Thanks for pinpointing out this and we have changed “level” to “phylum” (Supplemental Information, Fig. S2). The Cyanobacteria, namely the blue-green algae, is commonly found in natural environment like water. A possible explanation for detecting this organism in our study might be due to the food or water residue in the samples. This is not the only case specific to our data and we also observed Cyanobacteria in HMP2 released data (Lloyd-Price *et al.*, Nature 2019). However, the presence of this bacteria does not influence our main conclusions as its low relative abundance $< 0.05\%$.

The original reads are mapped to 19 phyla in total but we have removed those with extremely low-present bacteria after decontamination quality control (**Supplementary Methods**). We carefully compared our data with previous published studies like the one from Lloyd-Price *et al.*, and found that most of our detected bacteria are consistent with others. For example, the top most abundant taxa are Firmicutes, Bacteroidetes, and Proteobacteria which are also in line with the dominant bacteria reported in stool samples. To exemplify this, the following lists the bacteria detected in this study and Lloyd-Price *et al.* study.

Bacteria detected by 16S in this study
Acidobacteria, Actinobacteria, Bacteroidetes, Cyanobacteria, Deinococcus-Thermus, Epsilonbacteraeota, Euglenozoa, Firmicutes, Florideophycidae, Fusobacteria,

Gemmatimonadetes, Patescibacteria, Proteobacteria, Retaria, Spirochaetes, Synergistetes, Tenericutes, Thermotogae, Verrucomicrobia
Bacteria detected by 16S in Lloyd-Price et al study
Acidobacteria, Actinobacteria, Bacteroidetes, Cyanobacteria, Deinococcus-Thermus, Epsilonbacteraeota, Euglenozoa, Firmicutes, Florideophycidae, Fusobacteria, Gemmatimonadetes, Patescibacteria, Proteobacteria, Retaria, Spirochaetes, Synergistetes, Tenericutes, Thermotogae, Verrucomicrobia

(2.2) The authors need to use more specific language when describing changes in microbiota associated with CD vs UC biopsies in lines 168-174. I would also warn against broadly stating "overall mucosa-attached microbial composition was similar" when comparing at the genus level and ironically, in a subsection that's titled "composition is highly personalized".

Authors' reply: We agree that we could have been more precise in our wording in this particular section. Of course, we agree with the unintended contradiction introduced by this formulation and rephrased this sentence accordingly (lines 174-179).

"...Interestingly, across our cohort, few differentially abundant taxa were observed between colonic and ileal biopsies, and this appeared to be independent of inflammation. More specifically, only seven bacterial taxa were differentially abundant between patients and controls, which might however be driven by the relatively low number of non-IBD controls."

(2.3) I like the comparison-with-HMP2 snippets that the authors have included in response to Reviewer 1. Can the authors please clarify if they used all IBD samples or the subset IBD samples with active disease (dysbiosis score). Since inflammatory response is so central to this study, the authors should ideally compare their observations against the HMP2 inflamed and non-inflamed biopsies separately.

Authors' reply: We included all IBD samples from the HMP2 cohort and compared the microbial community composition mainly across CD, UC and non-IBD controls. We fully agree with the reviewer that ideally, we should compare our results with HMP2 in inflamed and non-inflamed biopsies separately. However, HMP2 only contained nine non-inflamed biopsies which limited our statistical comparison. Therefore, we emphasized that our comparison was restricted to a cross-disease groups and inter- or intra groups, but not extended to groups stratified by inflammation.

(2.4) The p-value is really small for the CD-UC comparison in Fig. 2C where the distributions look pretty similar. How many comparisons in each distribution and what test was used? Can the authors confirm with Bray-Curtis distance?

Authors' reply: This analysis aims to compare if the mucosal microbiota cross-disease group shows similar variation. In the original comparison, we calculated the Aitchison distance between all the biopsies from CD, UC and non-IBD control separately (n =75,855, n =46665 and n =1711). We also repeated the analysis using Bray-Curtis distance and similar observation is identified (mucosal microbiota of CD shows the largest variation).

All the p values from CD vs. control, UC vs. control and CD vs. UC are $< 2.2e-16$ and were derived from Wilcoxon tests.

(3) I am not quite sure how to interpret the HALLAgram (Fig. 3E). What do the authors mean by 'comparisons of patients with CD vs. UC' and how did the authors do the three diagnoses comparisons in the same model. I would also recommend using the most recent version of HALLA to reproduce these results.

Authors' reply: The 'comparisons of patients with CD vs. UC' here refers the sub-disease group comparison where the microbiota show differences between patients with CD and patients with UC. We have re-coded the diagnoses groups into dummy variables when using HALLA model. For example, when comparing CD vs. UC, we coded CD as 1 and UC as 0 while non-IBD control as NA; when comparing CD vs. control, we coded CD as 1 and control as 0 while UC as NA. We have used the most recent version of HALLA for our analysis.

(4) It would helpful to the reader to include a legend in Fig. 4A in addition to the details provided in the caption.

Authors' reply: We have added a legend in Fig. 4.

(5) There is a lot of white space in Fig. 5.

Authors' reply: We have adjusted the figure structure and removed the extra white space.

(6) Maintain consistency in figures for indicating 'non-IBD'. Some legends say "non-IBD" and others say "Control".

Authors' reply: We have changed "control" to "non-IBD" throughout all the figures.

(7) It is still not clear to me what the authors are trying to show in Fig. 7A. The y-axis needs to be more informative than "variation explanation" and the legend title is missing. Is the variation explained by something 60% for epithelial cells? What is that something? I urge the authors to add more details and explanation to the accompanying text.

Authors' reply: We apologize for the unclear interpretation of the analysis. In Fig. 7A, our goal is to demonstrate the extent to which basic factors, medication use, inflammatory status, tissue location and especially mucosal microbiota contribute to the degree of variation in expression of intestinal cell types (deconvoluted from RNA-seq data) (lines 393-402). The "variation explanation" refers to the total observed variance of a cell enrichment across the biopsies divided by the variance associated with the factors (e.g. bacteria and inflammation). The heatmap below this boxplot panel then designates what the relative contributions are from the different factors. In the example of epithelial cells, ~60% of the variation in epithelial cell type enrichment could be explained by the combination of displayed factors (basic factors (age, sex, and BMI), medication, inflammation, tissue location and mucosal microbiota). We have amended the accompanying text to clarify the purpose and meaning of this analysis (lines 393-413):

"These associations appeared evident within a combination of factors potentially contributing to the explained variation in intestinal cell type enrichment, including basic factors like age, sex, and BMI, as well as medication use, inflammatory status, and tissue location."

Moreover, we have modified the figure as reviewer suggested.

Reviewer #3 (Remarks to the Author):

The authors have satisfactorily addressed the concerns raised in the previous review. Additional analyses have been performed to address questions and the manuscript has been appropriately updated.

Authors' reply: We would like to thank the reviewer for the positive evaluation of our manuscript and the valuable comments that were provided for the revision of our manuscript. We are pleased to read that our revision has been well received and look forward to seeing our manuscript appearing online.